# Molecular pixelation: spatial proteomics of single cells by sequencing

Filip Karlsson [1] ✉, Tomasz Kallas[1], Divya Thiagarajan[1], Max Karlsson [1], Maud Schweitzer[1], Jose Fernandez Navarro [1], Louise Leijonancker[1], Sylvain Geny[1], Erik Pettersson[1], Jan Rhomberg-Kauert [1], Ludvig Larsson[1], Hanna van Ooijen[1], Stefan Petkov[1], Marcela González-Granillo[1], Jessica Bunz[1], Johan Dahlberg[1], Michele Simonetti [1], Prajakta Sathe[1], Petter Brodin [2,3,4], Alvaro Martinez Barrio[1] & Simon Fredriksson [1,5] ✉

The spatial distribution of cell surface proteins governs vital processes of the immune system such as intercellular communication and mobility. However, fluorescence microscopy has limited scalability in the multiplexing and throughput needed to drive spatial proteomics discoveries at subcellular level. We present Molecular Pixelation (MPX), an optics-free, DNA sequence-based method for spatial proteomics of single cells using antibody–oligonucleotide conjugates (AOCs) and DNA-based, nanometer-sized molecular pixels. The relative locations of AOCs are inferred by sequentially associating them into local neighborhoods using the sequence-unique DNA pixels, forming >1,000 spatially connected zones per cell in 3D. For each single cell, DNA-sequencing reads are computationally arranged into spatial proteomics networks for 76 proteins. By studying immune cell dynamics using spatial statistics on graph representations of the data, we identify known and new patterns of spatial organization of proteins on chemokine-stimulated T cells, highlighting the potential of MPX in defining cell states by the spatial arrangement of proteins.

The spatial organization of immune cell surface receptors governs multiple functions such as dynamic tuning of cell signaling[1], cell–cell communication[2], T cell effector function[3], movement via adhesion receptors[4], drug mode-of-action[5] and efficacy of cellular therapies[6]. Spatial protein organization has traditionally been studied with microscopy, using fluorophore-labeled antibodies on immobilized samples[7], or with imaging flow cytometry[8]. These typically provide data in one focal plane, in 2D, for about four targets per staining cycle[7]. For higher multiplexing in microscopy, iterative staining, imaging or bleaching is needed at the cost of sample throughput[7]. However, imaging flow cytometry cannot be subjected to imaging cycles. Signal to noise is also

hampered by autofluorescence and spectral bleedthrough between channels. Super-resolution imaging has provided groundbreaking insights into cellular functional responses and signaling[9], but is even more limited in multiplexing and throughput. Methods that rely solely on nucleic acid sequence to image biological samples have been proposed[10,11] and demonstrated for RNA in a diffusion-based mechanism[12]. These methods, also referred to as DNA microscopy, use multiple DNA tags to reflect biomolecule identity and position in the biological sample, and are anticipated to increase sample throughput, multiplexing and resolution, far beyond the limits of microscopy[10]. However, they have not yet been demonstrated for proteins.

[1]Pixelgen Technologies AB, Stockholm, Sweden. [2]Department of Women's and Children's Health, Karolinska Institutet, Solna, Sweden. [3]Department of Immunology and Inflammation, Imperial College London, London, UK. [4]Medical Research Council London Institute of Medical Sciences (LMS), Imperial College Hammersmith Campus, London, UK. [5]Royal Institute of Technology, Department of Protein Science, Stockholm, Sweden. ✉e-mail: filip.karlsson@pixelgen.com; simon.fredriksson@pixelgen.com

Here, we present MPX, which uses DNA-tagged AOCs bound to their protein targets on chemically fixed cells to survey cell surface protein arrangement in a highly multiplex fashion. The assay is performed without sample immobilization or single-cell compartmentalization, in a standard reaction tube. The spatial analysis of protein arrangement is enabled by serially forming two associations between spatially proximate AOCs into local neighborhoods through the incorporation of a unique molecular identifier (UMI), similar to the proximity barcoding assay[13]. The generated amplicons are sequenced and spatial relationships of proteins are inferred from graph representations of the data for each single cell.

AOCs bound to cells are associated into local neighborhoods using DNA pixels, which are single-stranded DNA molecules with a diameter of <100 nm (see Methods). The upper limit of resolution, 280 nm, was estimated by dividing the surface area of a lymphocyte by the average number of DNA pixels per cell (Supplementary Fig. 1). Each DNA pixel contains a concatemer of a UMI sequence called a unique pixel identifier (UPI) and is generated by rolling circle amplification from circular DNA templates. Once added to the reaction, each DNA pixel can hybridize to multiple AOC molecules in proximity on the cell surface. The UPI sequence of the hybridized DNA pixel is then incorporated onto the AOC oligonucleotide by a gap-fill ligation reaction, thus creating neighborhoods where the set of AOCs within each neighborhood share the same UPI sequence. Following enzymatic degradation of the first DNA pixel set, a second set of DNA pixels is similarly incorporated by hybridization and gap-fill ligation reactions (Fig. 1a). The generated amplicons are then amplified by PCR and sequenced. Each sequenced molecule contains four distinct DNA barcode motifs; a UMI to enable identification of unique AOC molecules, a protein identity barcode and two UPI barcodes with neighborhood memberships.

The relative location of each unique AOC molecule can be inferred from the overlap of UPI neighborhoods created from the two serial DNA pixel hybridization and gap-fill ligation steps (Fig. 1a). Each sequenced unique molecule can be represented as an edge in a bipartite graph, with UPI-A and UPI-B sequences as nodes and protein identity as edge attributes (Fig. 1b), or alternatively as a one-mode projected graph of UPI-A sequences as nodes and protein identities as node attributes. The graphs generated from a sequenced sample following data processing and filtering contain graph components that can be separated into distinct graphs corresponding to single cells. The spatial analysis of protein arrangement, such as the degree of clustering of a single protein or colocalization between two or more proteins, can be performed by interrogating the location of edge or node attributes on the graph representations of each cell.

Using a panel of 76 AOCs targeting immune cell surface proteins, and four control AOCs, we demonstrate the ability of MPX to generate single-cell data based on protein abundance from peripheral blood mononuclear cells (PBMCs). Next, the method was used to quantify the degree of spatial clustering or polarization from MPX polarity scores of each assayed protein upon modulation of the cell by a therapeutic antibody or by capping using a secondary antibody. Finally, abundance, polarity and pairwise colocalization of the target proteins was studied on immune cells subjected to chemotactic migration stimulation.

## Results

MPX data comprise both the spatial location and abundance of the targeted proteins, and can therefore be processed similarly to data from other single-cell technologies to annotate cells by their identity, as defined by the proteins displayed on their surface. To demonstrate the ability to generate single-cell data, a heterogeneous sample was processed using a 76-plex target panel against T cells, NK cells, B cells and monocytes, and the distribution of protein count signatures was assessed from the generated count data from each cell.

PBMCs from a healthy donor were fixed with paraformaldehyde (PFA) and assayed in two replicates by first staining with AOCs,

performing MPX, then subjecting a subset of 500 cells to PCR-based library preparation for sequencing. After data processing of sequence reads using the open source Pixelator pipeline (see Methods), 477 and 579 distinct cells were identified in the output data for the two replicates, which corresponded well with the 500 cells that were subjected to PCR and sequencing. The titration of cell input numbers between 50 and 1,000 has further shown that there is a strong correlation between the number of cells input to PCR and the number of detected cells in the data following Pixelator processing, indicating that one and not several connected graph components are generated from each cell after data processing (Supplementary Fig. 2). Graph connectivity is driven to a large degree by highly abundant and ubiquitous markers such as HLA-ABC, B2M and CD45, which are consistently abundant across various immune cell types.

The MPX protein count data matrices were processed to visualize and annotate cellular identities across the cells in the two replicate samples (see Methods; Data post-processing). Protein counts were centered log ratio (CLR)-transformed, and Louvain clustering identified seven clusters of cells with shared identity (Fig. 2a). The clusters were manually annotated using a differential abundance test (two-sided Wilcoxon rank-sum test, downsampled to 50 cells per cluster; Supplementary Table 1), in which each cluster was compared to other cells, resulting in 33 proteins with significantly different abundance (Bonferroni-adjusted $P < 0.01$, absolute average $\log_2(\text{fold change}) > 2$) (Fig. 2b). The cells were visualized in a uniform manifold approximation and projection (UMAP)[14] (Fig. 2a), which displays separated clusters corresponding to the main cell populations expected in PBMCs; T cells, B cells, NK cells and monocytes. The formation of separate cell type-specific clusters demonstrates that MPX is able to perform single-cell profiling without the need for physical compartmentalization of each individual cell or combinatorial cell barcoding using split-pool strategies.

The estimated frequencies of the identified cell populations were similar between the two replicate samples, with average levels around expected percentages; 70.0% T cells (52.5% CD4 T cells, 12.8% CD8 T cells, 4.6% mucosal-associated invariant T cells), 9.3% NK cells, 15.1% B cells and 5.6% monocytes (Fig. 2c). MPX generated on average 1,737 DNA-pixel A zones and 9,580 AOC UMIs per cell, with 5.6 UMIs per UPI-A pixel (Extended Data Fig. 1). Furthermore, cell population frequencies from MPX were consistent with those observed with flow cytometry, with similar but varied signal to noise for each marker (Supplementary Fig. 3 and Supplementary Table 2).

Abundance from all targets showed expected specificity patterns (Supplementary Fig. 4), exemplified by CD3 co-occurring with CD4 and CD8 in CD4 T cells and CD8 T cells, respectively, and by CD19 and CD20 exclusively found on B cells (Fig. 2d). Isotype control levels were low, with mIgG1, mIgG2a, and mIgG2b constituting only 0.03%, 0.15% and 0.04%, respectively, of the total AOC counts per cell on average (Supplementary Fig. 5). The cytoplasmic control target beta actin (ACTB), used to verify plasma membrane integrity, also showed low levels of 0.02%.

A technical duplet rate of less than 1% was estimated, as no cells with incompatible markers were found when mixing T and B cell lines fixed separately, followed by processing through the MPX workflow (Extended Data Fig. 2a,b). This is below the range observed for single-cell RNA sequencing methods[15] and also displays the specificity of the AOCs and of the MPX reaction. Additionally, the T and B cell mixing data were used to exemplify the dynamic range of individual AOCs expressed in only one of the two cell lines, with up to 100-fold difference in abundance levels for some markers (Extended Data Fig. 2c).

### Polarity scores by spatial autocorrelation from stimulated T cells and drug-treated B cells

The graph-based data generated by MPX can be used for spatial analysis by interrogating the edge or node attributes representing different

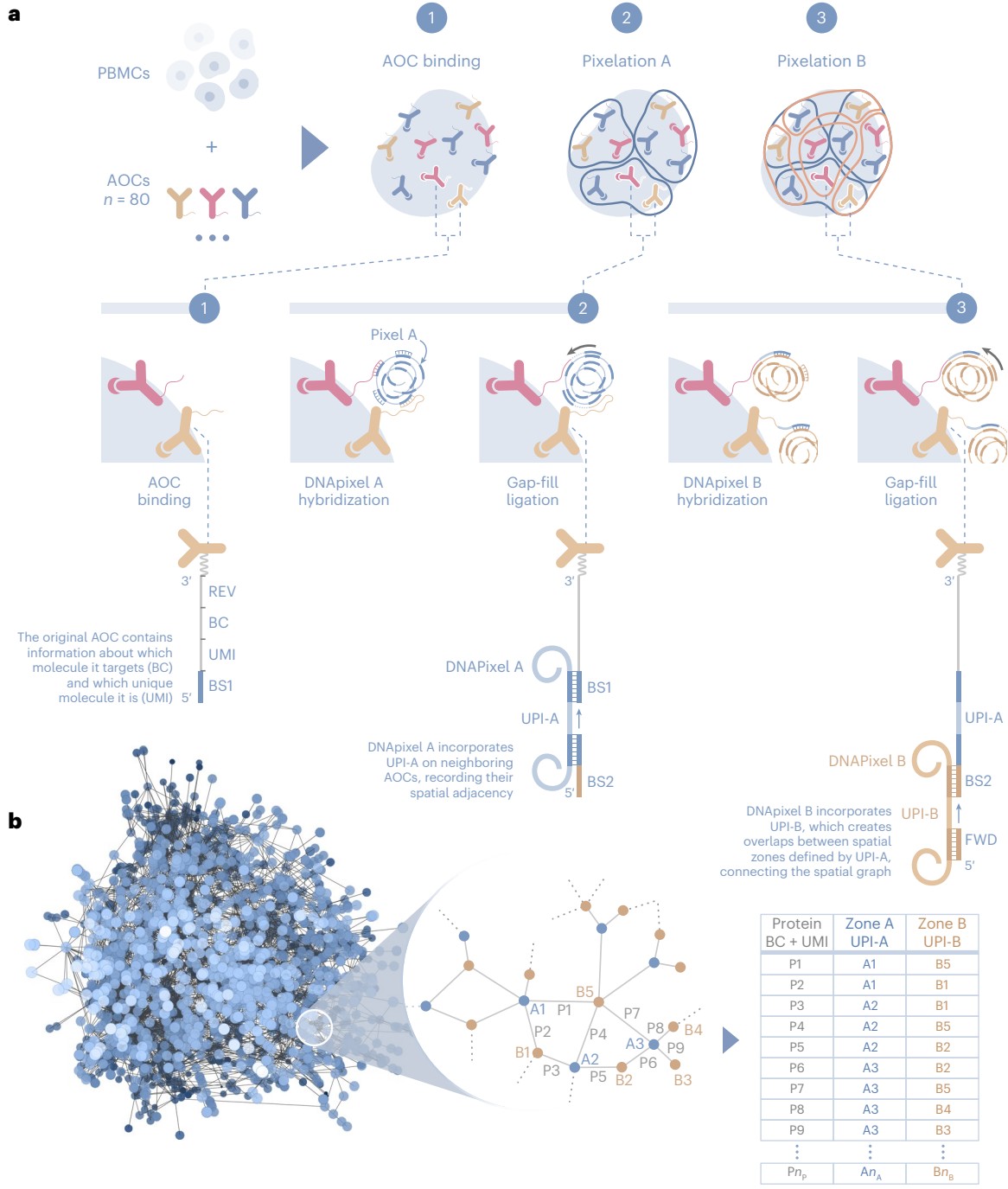

**Fig. 1 | Molecular Pixelation. a**, Barcoded AOCs are bound to their respective target proteins on cells (1). DNA pixel set A then hybridizes to the universal BS1 sequence of several proximal AOCs, followed by gap-fill ligation to incorporate the UPI sequence of each DNA pixel and a common pixel BS2, generating UPI-A-based spatial zones (2). Similarly, DNA pixel set B hybridizes to the extended AOCs, adding UPI-B (3). The final product is amplified by PCR using REV and FWD primer sites and sequenced. **b**, Each sequenced AOC molecule is represented as an edge in a bipartite graph, with the two UPI sequences as nodes and protein identities as edge attributes. The sequence data representing each cell generate a graph component from the association of AOCs into two partially overlapping UPI zones. The spatial analysis of protein arrangement is enabled by interrogating the location of node or edge attributes within the graph.

protein targets. Spatial autocorrelation can be used to measure clustering or nonrandomness of a spatial variable. A polarity score derived from the Moran's $I$ autocorrelation statistic was calculated for each protein marker per cell from spatial weights derived from the adjacency matrix of cell graphs (see Methods), in which positive polarity scores indicate clustered spatial distribution, and scores centered around zero indicate random spatial distribution.

To evaluate the ability to detect clustered protein expression from MPX data, we spatially clustered or 'polarized' CD3 by a capping reaction, using the CD3 AOC and a secondary anti-mouse antibody prior to PFA fixation, staining with the remaining AOCs and MPX. The PBMC MPX data were filtered for the T cell fraction, resulting in a total of 619 and 556 T cells for the capped and control samples, respectively. In a second approach to demonstrate protein polarization,

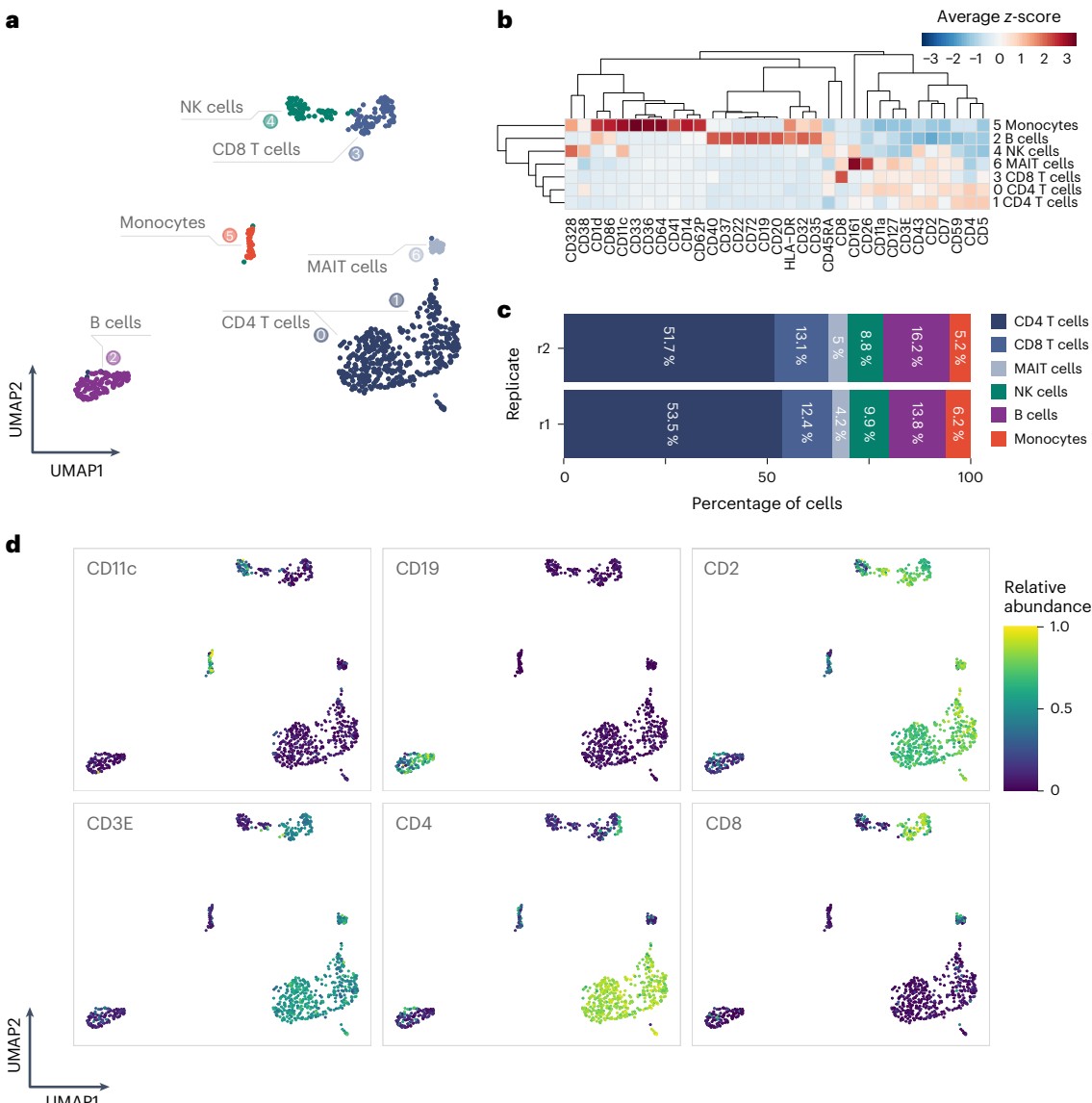

**Fig. 2 | Count data from an MPX experiment of healthy PBMCs. a**, UMAP of PBMCs, in which counts from each cell graph component are clustered into separated groups of the major cell types. The cluster numbers and cell type annotations are shown. MAIT, mucosal-associated invariant T cells. **b**, Heatmap with relative expression of differentially abundant proteins (Bonferroni-adjusted

$P < 0.01$ and absolute average $\log_2$(fold change) > 2), showing the average $z$-score for the cluster compared to other cells. **c**, Frequencies of annotated cell types per replicate. **d**, Relative abundance on CLR-transformed counts of canonical cell type markers.

we analyzed the distribution of CD20 after treatment of Raji cells with a rituximab-based AOC before PFA fixation. Rituximab, a monoclonal therapeutic antibody, is known to cluster CD20 on B cells, enhancing antibody-dependent cellular cytotoxicity-based cancer killing[5]. Untreated Raji cells were used as a negative control. A total of 684 and 440 Raji cells were included in the treated and control groups, respectively.

As seen in Fig. 3a,e, polarity scores were significantly elevated for the CD3-capped and the rituximab-treated samples (two-sided Wilcoxon rank-sum test, downsampled to 50 cells; Benjamini–Hochberg adjusted $P$ value of $4.2 \times 10^{-10}$ and $3.8 \times 10^{-14}$, respectively) compared to controls (Supplementary Table 3). No other proteins showed similar levels of difference between the treated sample and control, as can be seen in the resulting volcano plots (Supplementary Fig. 6). 2D graph representations (Fig. 3b,f) and spherical 3D density heat maps were generated from force-directed graph layouts of one representative CD3-capped T cell and one CD20–rituximab-treated

Raji cell to visualize the polarized distribution (Fig. 3c,g). Fluorescence microscopy for CD3 and rituximab was performed to validate the spatial redistribution of the target proteins upon treatment (Fig. 3d,h).

As an interesting note and to highlight the ability of MPX for new spatial discoveries through its strength of high multiplexing, we also found previously unreported strong clustering for CD54 and CD82 individually in Raji cancer B cells. This was not seen in PBMCs, which could potentially be exploited by cancer therapeutics based on antibody-dependent cellular cytotoxicity (Extended Data Fig. 3).

## Colocalization of protein pairs

The colocalization of cell surface proteins has a crucial role in driving cellular processes. To accurately quantify pairwise combinations of measured markers and identify patterns of colocalized protein groups, we developed an MPX colocalization score (see Methods). This score is based on Pearson's correlation coefficient ($r$) and Monte Carlo simulation, and enables comparison of measured scores with

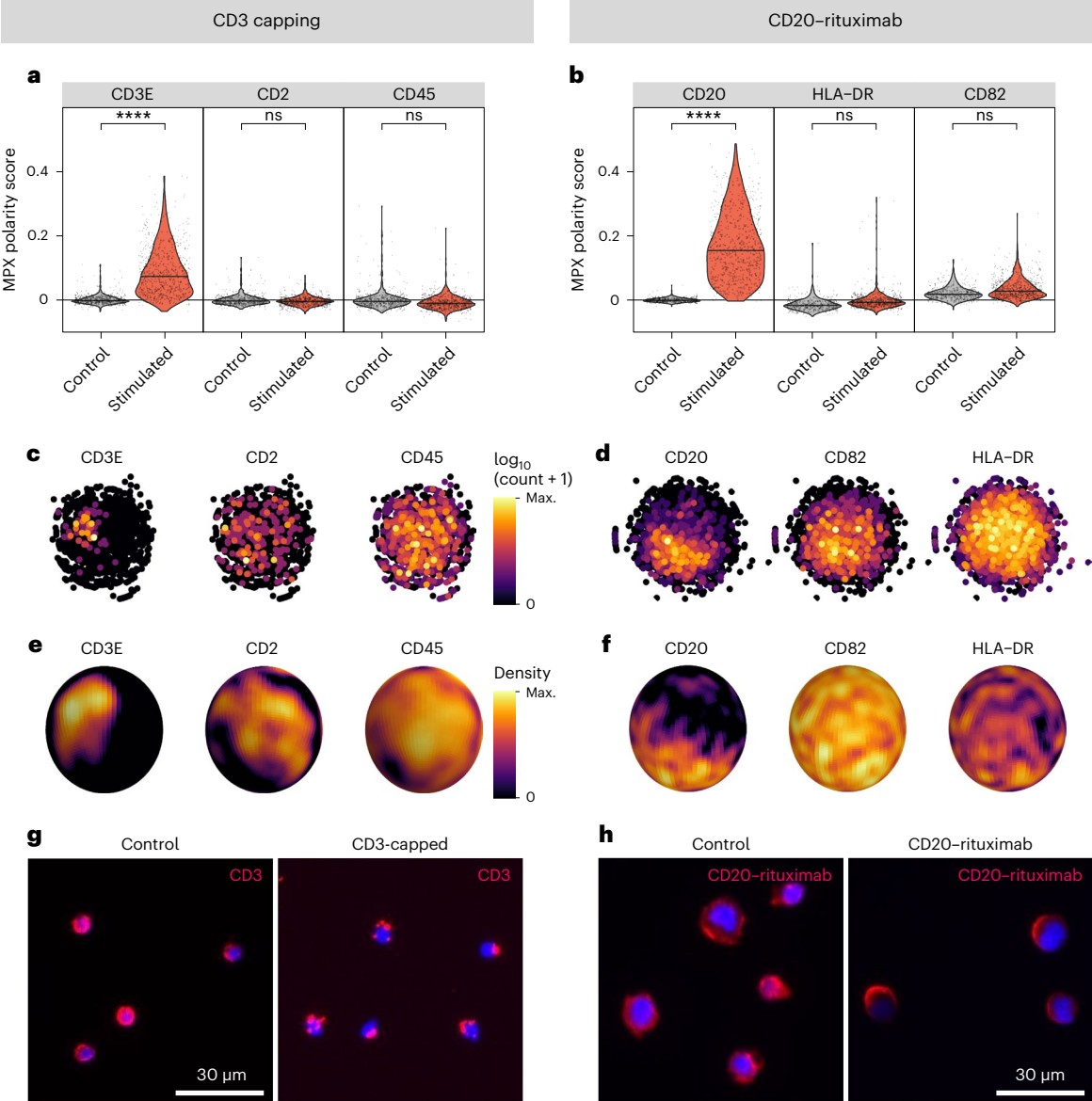

**Fig. 3 | Protein polarization triggered by cell stimulation detected by spatial MPX polarity scores for CD3-capped T cells and rituximab-treated Raji B cells. a,b,** Violin plot of polarity scores for CD3 and CD20–rituximab, and unaffected controls. Benjamini–Hochberg adjusted *P* values from two-sided Wilcoxon rank sum test. ****$P < 0.001$; ns, not significant. The adjusted *P* values are $4.24 \times 10^{-10}$ for CD3E (**a**) and $3.81 \times 10^{-14}$ for CD20 (**b**), whereas the adjusted *P* values of the other displayed markers are 1. **c,d,** 2D graph representations of MPX graphs of a CD3-capped cell (**c**) and a rituximab-treated cell (**d**) using Kamada–Kawai layout. Each point, representing an individual DNA pixel A, is colored in proportion to the log10(counts + 1) detected in the pixel. **e,f,** 3D density heat maps derived from the graph representations of a CD3-capped cell (**e**) and a rituximab-treated cell (**f**), colored by the count density of the indicated markers. **g,h,** Fluorescence microscopy validation of CD3-treated and rituximab-treated, and control samples.

scores obtained from simulated cells that possess an equal distribution of marker counts but random localization. The score reflects the degree of spatial co-occurrence of two proteins by quantifying the deviation from what would be expected by random chance (minimizing the influence of bias from experimental perturbations), as well as protein abundance.

A negative colocalization score may also provide valuable insight, and is a reflection of segregation of protein markers. One such example is found in interacting cells, such as B cell and T cell complexes. Further analysis of the PBMC dataset revealed a small fraction of cells with abundance signatures consistent with both B and T cells, indicating that they may be B–T cell complexes. Colocalization scores between multiple B cell-specific and T cell-specific marker pairs for these presumed complexes were strongly negative, illustrating the expected spatial segregation of these marker pairs. Furthermore,

graph visualization of the complexes showed a clear separation of the B and T cell markers into distinct regions in the graph (Extended Data Fig. 4a–f and Supplementary Video 1).

## Combined spatial analysis of chemotactic T cells

To further demonstrate the applicability of MPX, we detected cellular structures, namely uropod formation in migrating immune cells. Uropods are critical for cytotoxic T cells to infiltrate tumors[16], associated with immune checkpoint inhibition efficacy and overall cancer survival[17].

Two different types of conditions were applied to stimulate T cells to attain a migratory phenotype and form uropods. The cells were either in suspension, or immobilized to a plate coated with only CD54 (ICAM1) or in combination with chemokine stimulation by either CCL2 (MCP1) or CCL5 (RANTES).

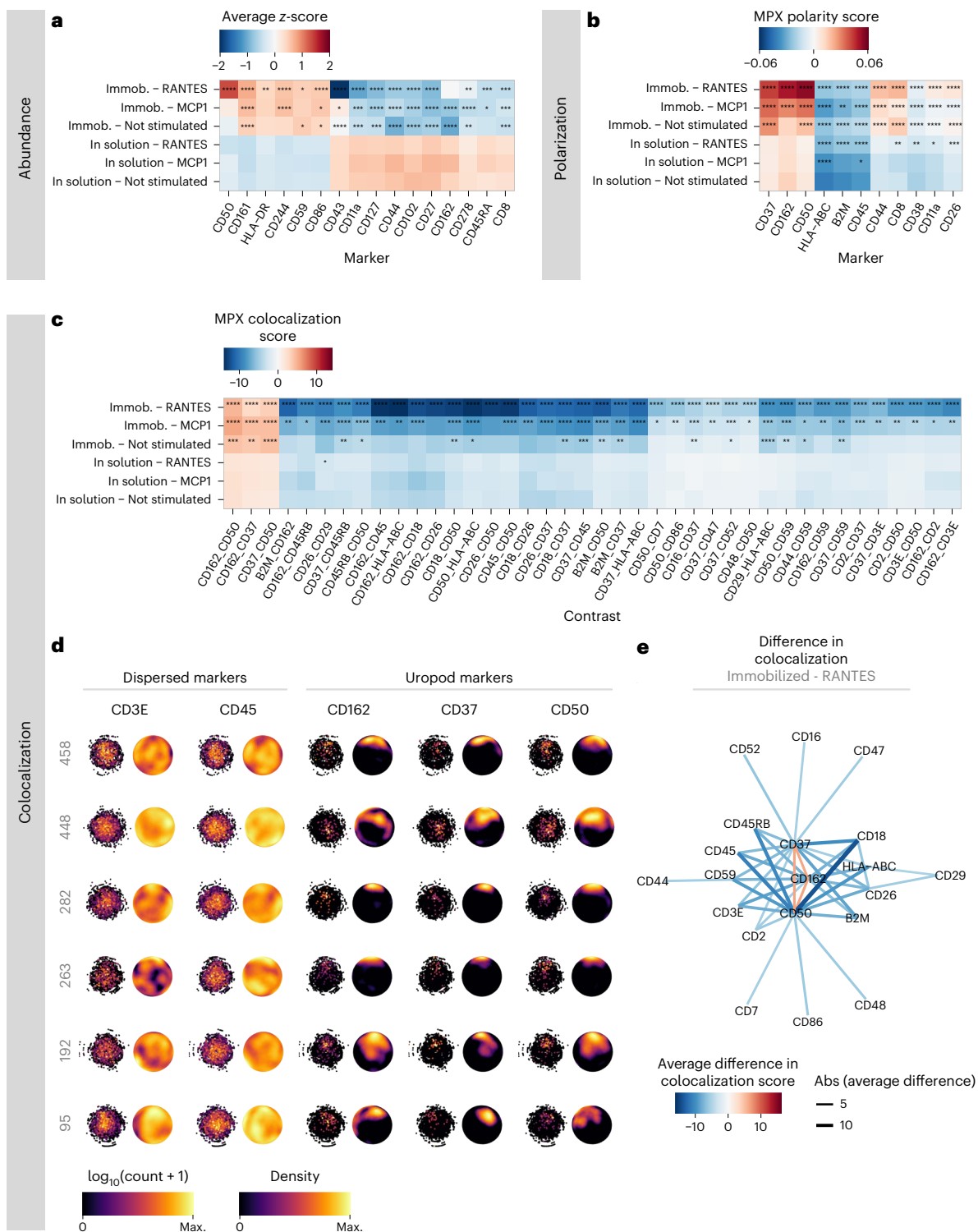

**Fig. 4 | MPX of chemokine-stimulated CD54 (ICAM1)-immobilized T cells.**
**a**–**c**, Heatmaps of the proteins with the largest effect size in differential analyses comparing abundance levels, polarity and colocalization. The statistical tests (two-sided Wilcoxon rank sum test) are performed by comparing each condition to the control sample of unstimulated T cells in suspension. The color of the tile encodes the mean $z$-score of CLR counts (**a**), the mean MPX polarity score (**b**) and mean colocalization scores (**c**). Benjamini–Hochberg adjusted $P$ values from two-sided Wilcoxon rank sum test. *$P < 0.05$, **$P < 0.01$, ***$P < 0.001$, ****$P < 0.001$. Immob, immobilized. **d**, Visualization of MPX graph components representing

six individual single cells (cell identifiers are denoted at the left of each row) showing two dispersed markers (CD3 and CD45) and three uropod markers (CD37, CD50 and CD162). Each column presents the 2D Kamada-Kawai graph layout of DNA pixels A (left) and the 3D density heat maps of the same cell (right). **e**, Graph representation of the relationship between differentially colocalized proteins from **c**. Protein pair links are colored by the average difference in colocalization scores in CD54-immobilized cells treated with RANTES compared to control.

The effects of the conditions were analyzed from three distinct perspectives: protein abundance, polarization and colocalization. The unstimulated in solution sample served as the reference condition, representing minimal uropod formation. Statistical significance was assessed using the Wilcoxon rank-sum test (two-sided) to determine whether there were significant differences in protein abundance, polarity scores and pairwise colocalization scores between each condition and the control sample (Supplementary Table 4). To test differences in abundance levels, cells were downsampled to 50 per condition for differential polarity and 100 per condition for colocalization.

The analysis of protein abundance and spatial distribution revealed notable differences compared to the reference condition. Interestingly, for nonimmobilized samples, no changes in protein abundance were seen upon stimulation, but significant changes were observed in the spatial metrics of polarization and colocalization (Fig. 4a–c and Extended Data Figs. 5–7). Effects on immobilized cells were more pronounced where the differential abundance analysis resulted in 16 proteins with significantly different protein abundance (Bonferroni-adjusted $P < 0.05$, average $\log_2$(fold change) > 0.3) in any of the conditions (Fig. 4a). The differential polarization analysis identified 11 markers exhibiting significant differentiation in protein polarization (Benjamini–Hochberg adjusted $P < 0.05$, average difference > 0.0125; Fig. 4b), and the differential colocalization analysis identified 40 marker pairs (corresponding to 20 unique proteins) displaying significant differences in protein colocalization (Benjamini–Hochberg adjusted $P < 0.05$, average difference > 5; Fig. 4c). This illustrates the importance of analyzing spatial relationships between proteins to gain insights into cellular states. Importantly, a notable overlap was observed between the markers demonstrating differential colocalization and those showing differential polarization, with eight proteins displaying significant differences in both spatial tests across at least one of the conditions. Among these proteins, CD50 and CD162, which are known to be located within the T cell uropod[18], exhibited differential spatial arrangement in CD54-immobilized and stimulated samples, with increased polarization and colocalization (Fig. 4b,c and Extended Data Fig. 8). 3D animations of a T cell displaying uropod formation are presented in Supplementary Video 2. Notably, CD37 has previously been described as participating in uropod formation in B cells, neutrophils and dendritic cells[19], and is here observed to increase polarization and colocalization with CD50 and CD162 upon CD54 immobilization and chemokine stimulation in T cells. Validation using fluorescent microscopy for CD50, CD162 and CD37 along with two non-polarized proteins (CD3 and CD45) confirmed the observed polarization patterns (Extended Data Fig. 9).

In addition, CD43 and CD11a, which are ligands of CD54 used in the stimulation or attachment of the T cells[18], showed decreased abundance and CD11a showed increased polarization, which may be effects of the CD54-mediated immobilization partially blocking or perturbing AOC binding. The polarization and pronounced three-way colocalization among CD50, CD37 and CD162 is visible in the graph representation of individual single cells in Fig. 4d (Supplementary Fig. 7). The three proteins (CD50, CD37 and CD162) simultaneously displayed decreased colocalization and segregation with 17 proteins, many of which are ubiquitous and dispersed in T cells, such as CD45, CD3E, HLA-ABC and B2M (Fig. 4e and Extended Data Fig. 10). This was expected, given that clustering of these three proteins led to a reduction in colocalization with other proteins.

These findings indicate that the MPX polarity score and MPX colocalization score show high concordance in detecting proteins with spatial rearrangement across conditions, while providing different yet complementary information about the spatial arrangement of proteins. The MPX polarity score indicates whether a protein is spatially clustered on the cell surface, whereas the colocalization score reveals the protein's spatial relationship with other proteins.

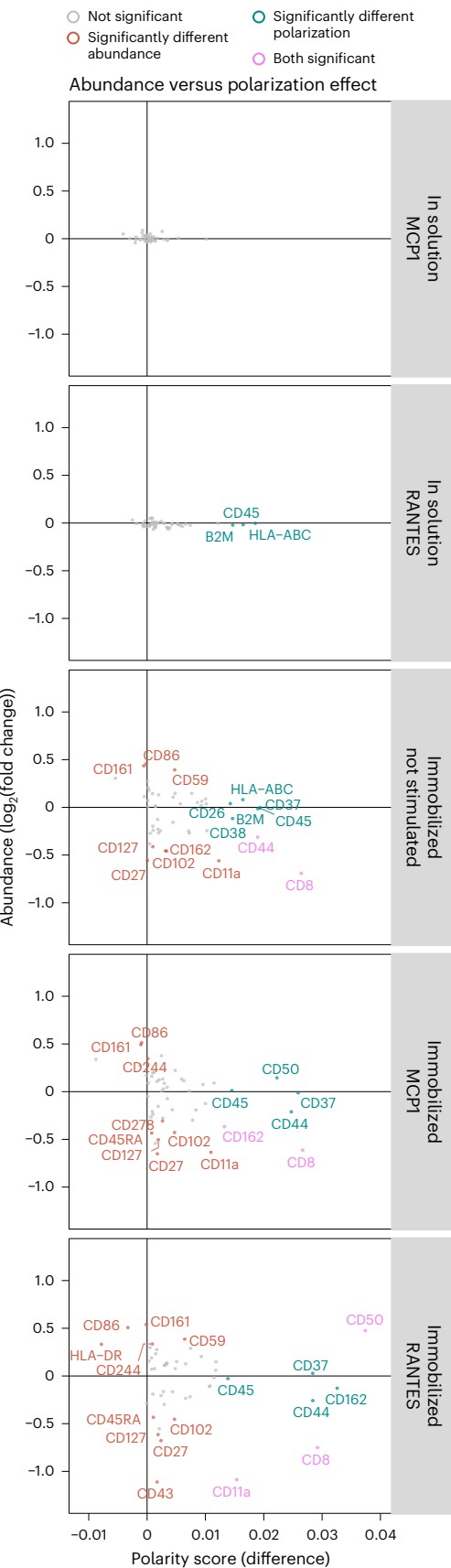

**Fig. 5 | The effect of T cell immobilization and chemokine stimulation.**
Comparison of the effects on target protein abundance (*y* axis) versus polarization (*x* axis) for each experimental condition compared to control. Abundance difference is expressed as $\log_2$(fold change) and the MPX polarity score as the average difference.

A spatial pattern of proteins indicating a uropod signal is expected to exhibit both increased polarization and colocalization, as observed prominently in CD37, CD50 and CD162, all of which are known proteins located in the uropod. This exemplifies how MPX can be used to identify patterns of protein spatial organization and their potential roles in cellular processes. In addition, the comparison of the effects of T cell immobilization on protein abundance (Fig. 4a) and protein polarization (Fig. 4b) revealed minimal overlap (Fig. 5) with merely three proteins (CD11a, CD8 and CD50) simultaneously showing significantly different abundance and polarization for the immobilized T cells treated with RANTES. These findings indicate a distinct and independent relationship between abundance and the two spatial metrics. Although we observed an effect on protein abundance, characterized by altered levels of uropod-associated proteins CD43, CD44, CD50, CD102 and CD162, the results are most pronounced in the colocalization data, highlighting the importance of incorporating spatial metrics of multiple proteins for a comprehensive and accurate understanding of cell states.

## Discussion

Attempts to increase multiplexing in fluorescence microscopy leads to lower sensitivity owing to spectral overlap issues, as well as lower throughput because of the need for multiple stain, wash, image and bleach cycles[7]. Converting proteins into DNA barcodes using proximity-dependent assays to increase multiplexing, sensitivity and throughput has been a successful strategy to enable large-scale, nonspatial plasma proteomics studies[20]. Using an optics-free DNA barcoding strategy also for spatial proteomics will drive similar advancements by the virtually endless supply of individually detectable sequence barcodes compared to available fluorophores, placing MPX on a promising trajectory for even further development. The potential of microscopy-by-sequencing over traditional imaging has been described in a review as enormous[10].

To the best of our knowledge, MPX is a new method to identify the relative locations of a large number of proteins in single cells without the use of fluorescence microscopy. The method enables a unique combination of multiplexing, throughput and spatial resolution in 3D, providing spatial dimensions to proteomics-level single-cell research. This field holds potential for new insights into essential cellular activities such as cell motility, cell activation, drug mode-of-action, drug-target discovery based on spatial clustering and formation of cell–cell communication interfaces. The uropod formation study presented here highlights the importance of the spatial proteomics of single cells, as the data relating to the localization of multiple proteins most strongly reflect the migratory status of T cells compared to abundance measurements. MPX could guide the development of therapies that aim to spatially reorganize cell surface proteins in order to support immunomodulatory activities[21,22], including pretuning immune synapses in chimeric antigen receptor T cells[23].

MPX enables the representation of cellular states as graph objects. Spatial statistics, including permutation testing, can then be leveraged to enable the identification and quantification of cellular changes relative to randomness. In addition, graph data are particularly well suited for neural networks and machine learning, which are increasingly influential in the realm of bioinformatics and computational biology[24]. The laboratory protocol takes 2 days and requires 120,000 reads per cell to ensure robust spatial maps. Throughput is expected to scale with sequencing capacity. We anticipate new algorithm development to go beyond the colocalization of protein pairs, to the analysis of spatial constellations of multiple proteins. The uniquely combined features of multiplexing, throughput and data analysis position MPX as a promising new method for spatial proteomic discoveries that can be implemented in any laboratory without dedicated instrumentation. A notable limitation of MPX is the difficulty to precisely assign a spatial resolution owing to the polymer nature of the molecular DNA pixels.

Also, in areas in which targeted proteins are not present, a negative spatial signal is not generated as in light-based microscopy.

As illustrated by the uropod formation study, high multiplexing with spatial dimensions in 3D can provide insights into the molecular mechanisms behind T cell motility, which is essential in the lymphocyte infiltration of solid tumors during immune therapies. As MPX translates proteins into nucleic acid-based spatial graphs, we anticipate that its applications will extend to facilitating the analysis of other biomolecules such as RNA, increasing multiplexing capacity and enabling the analysis of other sample types such as FFPE tissues (currently in progress). MPX will benefit from the rapid advances in DNA sequencing, computational power, algorithm development and machine learning.

## Online content

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

## Methods

### PBMC extraction from whole blood

PBMCs were separated from whole blood collected in heparin or EDTA blood collection tubes by Ficoll-Paque density gradient centrifugation. The platelet fraction was reduced by three repeated centrifugation steps at $100 \times g$ for 10 min.

### CD3 capping of PBMCs

PBMCs were first incubated with 50 µg ml$^{-1}$ of human IgG for 15 min at 4 °C to block Fc receptors and then incubated with 20 µg ml$^{-1}$ of anti-CD3 AOC for 40 min at 4 °C. After two washes, cells were incubated with 40 µg ml$^{-1}$ of goat anti-mouse secondary antibody for 40 min at 4 °C, followed by incubation at 37 °C for 1 h. The capped cells were then immediately fixed with PFA.

### Rituximab treatment of Raji cells

Raji cells were Fc-receptor blocked with 50 µg ml$^{-1}$ of human IgG for 15 min at 4 °C and washed. Cells were then either PFA-fixed directly or incubated with 20 µg ml$^{-1}$ of rituximab AOC in RPMI medium for 60 min at 37 °C, followed by washing and PFA fixation.

### Microscopy validation

Fluorescence microscopy validation was performed on a Leica DM RB microscope with a ×20 objective, using a Leica K5 sCMOS camera after incubation of 40 µg ml$^{-1}$ of goat anti-mouse secondary antibody carrying phycoerythrin fluorophore for 40 min at 4 °C, followed by incubation with Hoechst for 5 min at room temperature for nuclei staining.

### Uropod formation of T cells

PBMCs were first stimulated into PHA blasts with PHA-L for 48 h, washed twice in PBS and incubated with 10 ng ml$^{-1}$ of IL-2 in RPMI medium for 5 days at 37 °C. Cell culture plates were coated with either 5 µg ml$^{-1}$ of CD54Fc antibody alone or together with either 10 ng ml$^{-1}$ of RANTES or MCP1 at 37 °C for 1 h. Approximately 300,000 PHA blasts were added to each of the three coated plates for 2 h. The adhered cells were then fixed with 1% PFA, before being brought back into suspension through incubation with TrypLE enzyme solution (Thermo Fisher Scientific) for 10 min at 37 °C. For the solution conditions, PHA blasts were instead incubated in solution with either 10 ng ml$^{-1}$ of RANTES or MCP1, followed by PFA fixation.

### Cell fixation and AOC staining

Cells resuspended in PBS were fixed for 15 min at room temperature in a fixation solution containing 1% v/v PFA in PBS. Cells were washed once in PBS, followed by addition of a blocking or quenching buffer containing 1% FBS, 0.1% BSA, 1 mg ml$^{-1}$ of single-stranded DNA, 50 µg ml$^{-1}$ of human IgG, 125 mM of glycine, 4 mM of EDTA and 0.04% ProClin 300 in PBS. Cells were incubated for 15 min at 4 °C, followed by a wash in PBS to remove the blocking or quenching solution.

Fixated and blocked cells were stained for 30 min at 4 °C in a 50 µl reaction containing a cocktail of 80 AOCs, each at a concentration of 5 µg ml$^{-1}$, in a staining buffer comprising 0.2% BSA and 2 mM of EDTA in PBS. After three washing steps in wash buffer (50 mM of NaCl, 1 mM of EDTA, 20 mM of Tris-HCl, pH 8), AOCs bound to cells were stabilized using a secondary antibody by incubating the cells for 30 min at 37 °C in a secondary antibody solution consisting of 20 µg ml$^{-1}$ of secondary antibody[25], 0.2% BSA and 2 mM of EDTA in PBS, followed by two washing steps in wash buffer, before proceeding with the MPX protocol.

### MPX protocol

DNA pixels were allowed to hybridize to AOCs on 15,000 stained cells in a 50 µl A-pixel hybridization reaction containing 1 nM of A pixels, 2 µM of A-aap-fill polymerization (GFP) oligonucleotide and 1 µg µl$^{-1}$ of sheared salmon sperm DNA, in a hybridization buffer containing 300 mM of NaCl, 15 mM of MgCl$_2$, 20 mM of Tris-HCl (pH 8) and 0.05% Tween20. The reaction was incubated for 15 min at 55 °C, followed by two washes in wash buffer.

A-pixel UPIs were incorporated onto the monoclonal antibody oligonucleotides via a 50 µl gap-fill ligation reaction consisting of 40 U of Taq ligase (New England Biolabs), 3 U of T4 DNA polymerase (New England Biolabs), 100 µM of deoxynucleotide triphosphates (dNTPs) and 0.5 mM of NAD+ in 1× rCutSmart buffer (New England Biolabs). The gap-fill reaction was incubated at 37 °C for 20 min, followed by one wash in wash buffer.

A pixels were degraded by incubating the cells in a 50 µl reaction containing 1 U of USER enzyme (New England Biolabs) in wash buffer. The reaction was incubated at 37 °C for 30 min, followed by one wash in wash buffer.

Hybridization of B pixels was performed in the same reaction conditions as described for A pixels with the exception of using B-GFP oligonucleotide instead of A-GFP. Similarly, the gap-fill ligation reaction for B pixels was performed in the same reaction conditions as described for A pixels.

To remove any incomplete assay products, an exonuclease treatment targeting DNA with unprotected 5′ ends was performed. Cells were first counted using a hemocytometer, and an aliquot of less than 1,000 cells was resuspended into a 15 µl reaction containing 10 U of lambda exonuclease, 100 µM of dNTPs, 1 mM of NAD+ in 1× rCutSmart buffer (New England Biolabs). The reaction was incubated at 37 °C for 30 min, followed by heat inactivation at 75 °C for 10 min. See Supplementary Protocol for more information.

### PCR and next-generation sequencing

PCR was performed in a 40 µl reaction containing 1× Q5 HotStart Hifi PCR master mix (New England Biolabs), 0.4 µM of Illumina adapter PCR primers (ILM_p5_PCR, ILM_p7_PCR) containing eight nucleotide sample indexes to allow multiplexing and 15 µl of sample from the lambda exonuclease step.

The PCR products were purified twice using AmpureXP SPRI beads (Beckman Coulter) according to the manufacturer's instructions and subsequently quantified using Qubit HsDNA assay (Thermo Fisher Scientific). The purified PCR products were diluted to 0.65 nM with 15% phiX spiked in and paired-end sequenced on an Illumina NextSeq2000 sequencing system, using 44 cycles for Read 1 and 78 cycles for Read 2.

### AOC preparation

Monoclonal antibody clones (https://antibodies-online.com), prevalidated for specificity by the supplier and in numerous publications, were again validated in this study one by one for specific target binding on PFA-fixed PBMCs, using flow cytometry. Then they were coupled to oligonucleotides using DBCO-Azide click-chemistry[26] (Supplementary Table 5). After oligonucleotide coupling, each AOC was validated using MPX on multiple cell types with known positive or negative expression to confirm both functionality and specificity after conjugation. For convenience, all antibodies and AOCs were validated and used under the same conditions, including concentration.

### Rolling circle amplification template preparation

Circularized DNA templates were prepared by incubating 300 nM of template oligonucleotides with 200 nM of padlock probe in a 50 µl ligation reaction containing 1 mM of ATP and 400 U of T4 DNA ligase (New England Biolabs) in a buffer comprising 33 mM of Tris-acetate (pH 7.9), 10 mM of magnesium acetate, 66 mM of potassium acetate, 0.1% Tween20 and 1 mM of dithiothreitol. The reaction was incubated for 30 min at 37 °C, followed by heat inactivation at 75 °C for 10 min. Then, 10 U of Exonuclease I and 20 U of Exonuclease III (New England Biolabs) were added to each ligation reaction, which were incubated at 37 °C for 30 min, followed by heat inactivation at 85 °C for 20 min (see Supplementary Table 1 for oligonucleotide sequences).

## DNA pixel preparation

DNA pixels were prepared in 75 µl rolling circle amplification (RCA) reactions comprising 5 nM of circularized RCA template, 7.5 U of phi29 enzyme (Thermo Fisher Scientific), 0.75 mM of dAUGC or dNTPS, in the same reaction buffer as the one used for RCA template preparation. The reactions were incubated at 37 °C for 4 min, followed by heat inactivation at 65 °C for 10 min. rSAP enzyme (1 U) was added to each reaction to inactivate free dAUGCs or dNTPS, and the reactions were then incubated at 37 °C for 20 min, and then at 65 °C for 5 min (see Supplementary Table 2 for oligonucleotide sequences). DNA pixels were analyzed by scanning electron microscopy and shown to be spherical molecules below 100 nm in diameter (Supplementary Fig. 1).

## Data analysis using Pixelator

MPX sequencing data were analyzed using a dedicated open-source data analysis pipeline, Pixelator. First, sequence reads were quality-filtered to remove low-quality reads. Next, reads were matched against the common pixel binding sequence motifs (BS1 and BS2), and reads with a mismatch of >10% were discarded. The BS1 and BS2 sequences were then discarded, and only the UMI, BC, UPI-A and UPI-B sequence motifs were kept for each read. Duplicate reads generated from the PCR step before sequencing were collapsed into unique sequences defined by the combination of the 10 nt UMI and the 25 nt UPI-A sequences. The correction of PCR and sequencing errors was performed by clustering the set of putative unique UMI–UPI-A sequences with the help of a modified approximate nearest neighbor algorithm and by identifying the most frequent sequence from the read sequences that clustered together. The UPI-A, UPI-B, UMI and BC sequence motifs were extracted from each unique read and stored as an edge list. An undirected graph was generated from the UPI-A and UPI-B sequences of the edge list from a sequenced sample.

Community detection based on modularity maximization was performed on the resulting graph with the Leiden algorithm to identify and remove any spurious edges connecting otherwise densely connected communities assumed to represent cells. The resulting graph contained a set of connected components, each being a disjoint subset of the graph not connected to any other connected components. Component memberships were assigned to each edge of the edge list, and a count matrix was generated by summing up counts per component for each protein of the edge list. CLR-transformed counts were calculated per component member and used for cell annotation purposes.

From the total number of edges of each component membership, size outliers were identified based on the descending rank order distribution. A size threshold based on the rate of change was defined by finding the first and second derivatives from a univariate smooth spline curve fitted to the linear-log distribution of the ranked antibody count data. Edges for components considered as size outliers were removed from the edge list. The components that remained after size filtering were considered to represent cells.

Finally, the filtered edge list was used for downstream analysis, such as calculation of MPX polarity scores and colocalization scores, by interrogating the UPI-A one-mode projected graph from each cell. The graph generated from the UPI-A and UPI-B sequences of the edge list is bipartite, forming edges only between UPI-A and UPI-B nodes, and never between two UPI-As or two UPI-Bs. The one-mode projections of the graphs, with edges linking directly between UPI-A nodes, and the set of antibody counts associated with each UPI-A node as node attributes were used for calculating spatial metrics such as MPX polarity scores and colocalization scores.

## MPX polarity score calculation

Spatial autocorrelation was used to quantify the degree of clustering or nonrandomness of the spatial locations of each protein within each UPI-A one-mode projected cell graph. Moran's index ($I$) of spatial autocorrelation was computed for each CLR-transformed protein node count from every cell graph based on the spatial weights defined by the row-normalized adjacency matrix of the one-mode projected cell graphs. In addition to Moran's $I$ value, a $z$-score and a $P$ value were computed under the randomization null hypothesis for each cell graph. The obtained Moran's $I$ was referred to as 'MPX polarity score', showing the level of clustering or nonrandomness of each protein in the context of the spatial structure of the cell graph that they were calculated from. Elevated scores would be indicative of clustering or nonrandomness, and scores centered around zero would suggest a spatial pattern not significantly different from randomness.

## MPX colocalization score calculation

The calculation of 'MPX colocalization scores' per pairwise combination of proteins consisted of six steps, performed for each individual UPI-A one-mode projected cell graph. The first step involved stringent filtering on the protein counts, which removed data points with few counts to generate a robust score, disregarding markers with less than ten counts in total. The second step involved neighborhood aggregation, in which counts from A pixels in immediate connected neighborhood nodes were summed up, disregarding spatial neighborhoods with less than five counts. The third step was permutation, which generated simulated components with permuted protein localization. These components were used to create a null distribution that enabled the calculation of how much the resulting score deviated from random colocalization. The fourth step involved log-transformation (log1p), which was applied to the count data of the original component as well as to the permuted components. The fifth step involved calculation of the colocalization score statistic, Pearson's $r$. Last, the sixth step was null distribution fitting, which compared the observed score to the scores of the permuted components to generate a $P$ value and $z$-score that describe the degree to which the score was lower or higher than what was expected by chance.

## Graph visualization in 3D

Three approaches were used to generate graph layouts for visualization purposes. It is important to note that all data analyses were performed on raw graphs and not on layouts or spherical representations. In the first approach, Euclidean coordinates in 3D were generated for each node by applying the Kamada–Kawai force-directed graph layout algorithm on the graph representation of a cell. For the second approach, the generated coordinates were projected onto the unit sphere by dividing each coordinate by its norm. The subset of graph nodes associated with a protein marker of interest was colored.

For the third approach, a density heatmap was generated from the sphere-projected coordinates. A color value representing count density was obtained for each point of a 3D grid representing the surface of the unit sphere. This was achieved by applying a function that iterated over each surface grid point and calculated the distance, defined by the Euclidean norm, to each of the sphere-projected node coordinates associated with a specific protein target. For all node coordinates within a selected distance cutoff value to each grid point, $1 -$ distance/distance_cutoff was calculated, and the logarithm of this sum was used as the color value for each grid point.

## Data post-processing

The cells were filtered to remove size outliers, a common practice in flow cytometry and single-cell analysis, removing the ten largest components and setting a manual cutoff for the minimum number of detected UMIs per cell. This cutoff was set by examining the distribution of the number of UMIs in an 'edge rank plot' (Supplementary Fig. 8) showing the number of UMIs per component versus its rank, and selecting a cutoff approximating the elbow point where the component size is sharply declining in relation to the size rank.

The minimum component size cutoffs selected per dataset were 4,000 for the unstimulated PBMC experiment, 5,000 for the CD3 capping experiment, 20,000 for the Raji experiment and 8,000 for the uropod experiment. Next, the cells were filtered for outliers in terms of distribution of counts across AOCs using the Tau metric[27], a numerical value between 0 and 1 describing the degree to which the distribution of counts is skewed across markers. Even distribution across markers generates a Tau of 0, and a distribution completely skewed to a single marker results in a Tau of 1 (see Methods). Antibody aggregates often consist of a random composition of antibodies (high complexity) or are composed of a single antibody (low complexity), resulting in a notably low and high Tau score, respectively. The Pixelator pipeline marks components as 'High Tau' if the Tau score is above 0.995 or deviates from the population median with more than two interquartile ranges (IQRs), and as 'Low Tau' if the Tau score deviates more than 5 IQRs from the population median. Downstream analysis was performed using Seurat (v.4.3.0)[28] in R (v.4.2.2), as well as broom (v.1.0.2), rstatix (v.0.7.1), SeuratObject (v.4.1.3), tidygraph (v.1.2.2) and tidyverse (v.1.3.2)[29] for analysis, and ggforce (v.0.4.1), ggplot2 (v.3.4.0)[30], ggplotify (v.0.1.0), ggpubr (v.0.5.0), ggraph (v.2.1.0), graphlayouts (v.0.8.4)[31], igraph (v.1.3.5)[32], patchwork (v.1.1.2), pheatmap (v.1.0.12), plotly (v.4.10.1)[33], viridis (v.0.6.2) and viridisLite (v.0.4.1) for data visualization. Count data were CLR-transformed over each cell, and protein counts were scaled per protein by their mean and s.d. centered before dimensionality reduction and clustering. UMAPs were created from the first 12 components of a principal component analysis performed using markers, but excluding platelet markers (CD9, CD29, CD36, CD41, CD62P) and control markers (ACTB, mIgG1, mIgG2a, mIgG2b). Parameters included n_neighbors = 25, spread = 2 and min.dist = 0.1. Cell clustering was performed by Louvain community detection on a shared nearest neighbor graph at default settings (k.param = 20, prune.SNN = 1/15).

### T cell gating (CD3 polarization analysis)
The T cell fraction of the PBMC sample was extracted by applying cutoffs on CLR-transformed counts for T cell markers of >0.9 for CD3 and either >1.8 for CD4 or >1.5 for CD8. Cells other than T cells were removed from the data by filtering the cell-count matrix at <0.5 CD19, <1.1 CD20 and <0.5 CD14. The CLR-transformed count distributions and the cutoffs used to filter the data are shown in Supplementary Fig. 9.

### Doublet estimation
To estimate the rate of doublets, Daudi and Jurkat cells were fixed separately, then counted and mixed at a 50:50 ratio before proceeding with AOC staining and the MPX protocol.

### Reporting summary
Further information on research design is available in the Nature Portfolio Reporting Summary linked to this article.

### Data availability
The MPX raw read data and Pixelator 0.12 processed output can be downloaded from Pixelgen Technologies (https://software.pixelgen.com/datasets). Datasets are granted under a Creative Commons Attribution (https://creativecommons.org/licenses/by/4.0/legalcode) license. Source data are provided with this paper.

### Code availability
All analysis code can be found at https://github.com/Pixelgen-Technologies/pixelgen-MPX-paper.
A nf-core pipeline to run Pixelator on MPX data with demo examples is available at nf-core (https://github.com/nf-core/pixelator). A python package for MPX data analysis, data processing and algorithms on Pixelator is available through Pixelgen Technologies Github (https://github.com/PixelgenTechnologies/pixelator).

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

### Acknowledgements
This work was supported by the Wellcome Leap ΔTissue Program and Stiftelsen för Strategisk Forskning (SSF). P.B. is supported by a proof-of-concept grant from the Knut & Alice Wallenberg Foundation (grant no. 2023-0195).

### Author contributions
F.K. and S.F. conceived the Molecular Pixelation method. F.K., S.F., T.K., D.T., L.L., S.G., M.K., E.P., J.B., M.G.G., J.R.K., J.D., M. Schweitzer, M. Simonetti, H.v.O. and P.S. designed and/or performed experiments validating the methods and developed the specific reagents needed. F.K., A.M.B., L.L., S.P., J.F.N., J.D., M.K. and J.R.K. developed the data analysis algorithms and the pipeline. F.K., M.K., A.M.B., P.B. and D.T. performed in-depth analyses of biological results. S.F., F.K., M.K., P.B. and A.M.B. wrote the paper. All authors have read and approved the final paper.

### Funding

### Competing interests
All authors are employees or advisors to Pixelgen Technologies, which commercializes products based on Molecular Pixelation. They are also shareholders or stock option holders of the company.

### Ethics
All PBMC samples were purchased from a Karolinska Hospital blood bank drawn from healthy volunteers with informed consent, and sample identity and other medical information were withheld.

### Additional information
**Extended data** is available for this paper at https://doi.org/10.1038/s41592-024-02268-9.

**Correspondence and requests for materials** should be addressed to
Filip Karlsson or Simon Fredriksson.

**Peer review information** *Nature Methods* thanks the anonymous
reviewers for their contribution to the peer review of this work.

Primary Handling Editor: Madhura Mukhopadhyay, in collaboration
with the *Nature Methods* team. Peer reviewer reports are available.

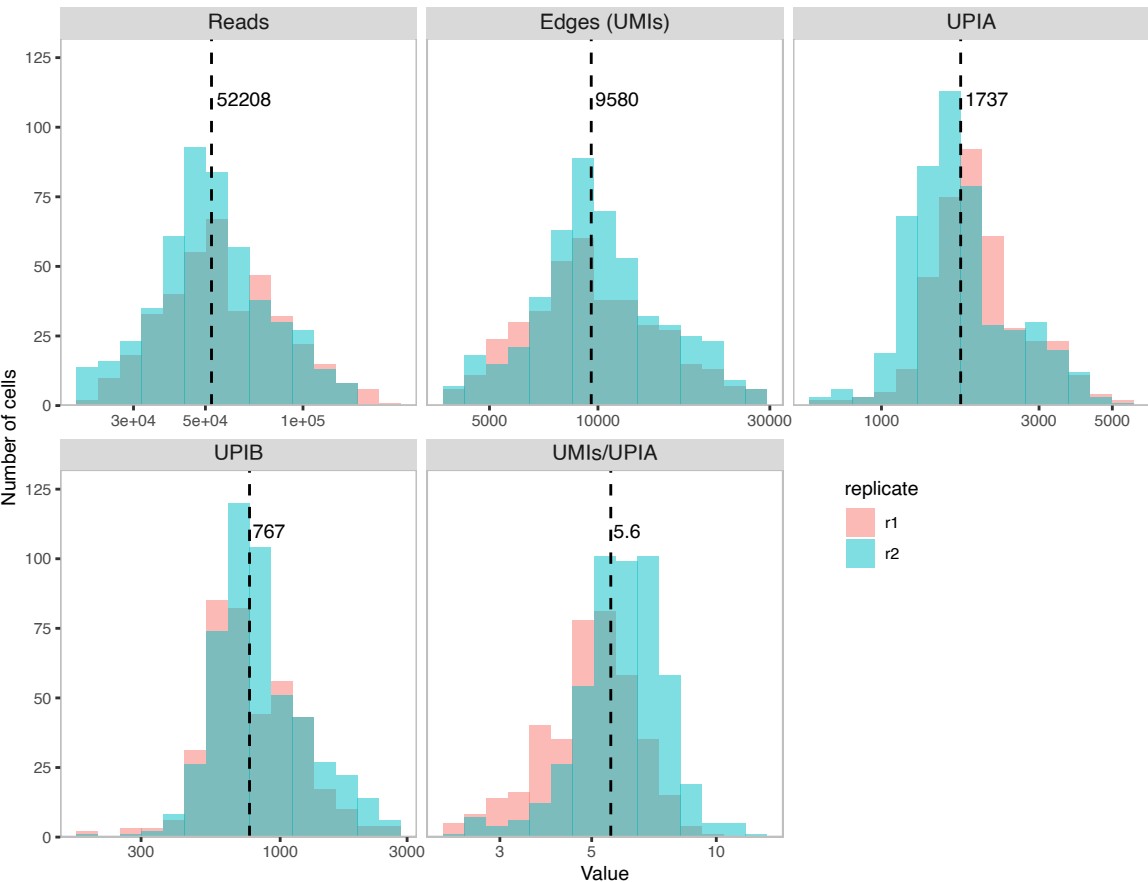

**Extended Data Fig. 1 | Distribution of metrics per component (single cell).** Histograms showing the distribution of the number of reads, edges (UMIs), DNA-pixels A (UPI-A), DNA-pixels B (UPI-B), and the number of unique molecules per DNA-pixel A (UMIs/UPI-A). The median of each metric is marked by a dashed line.

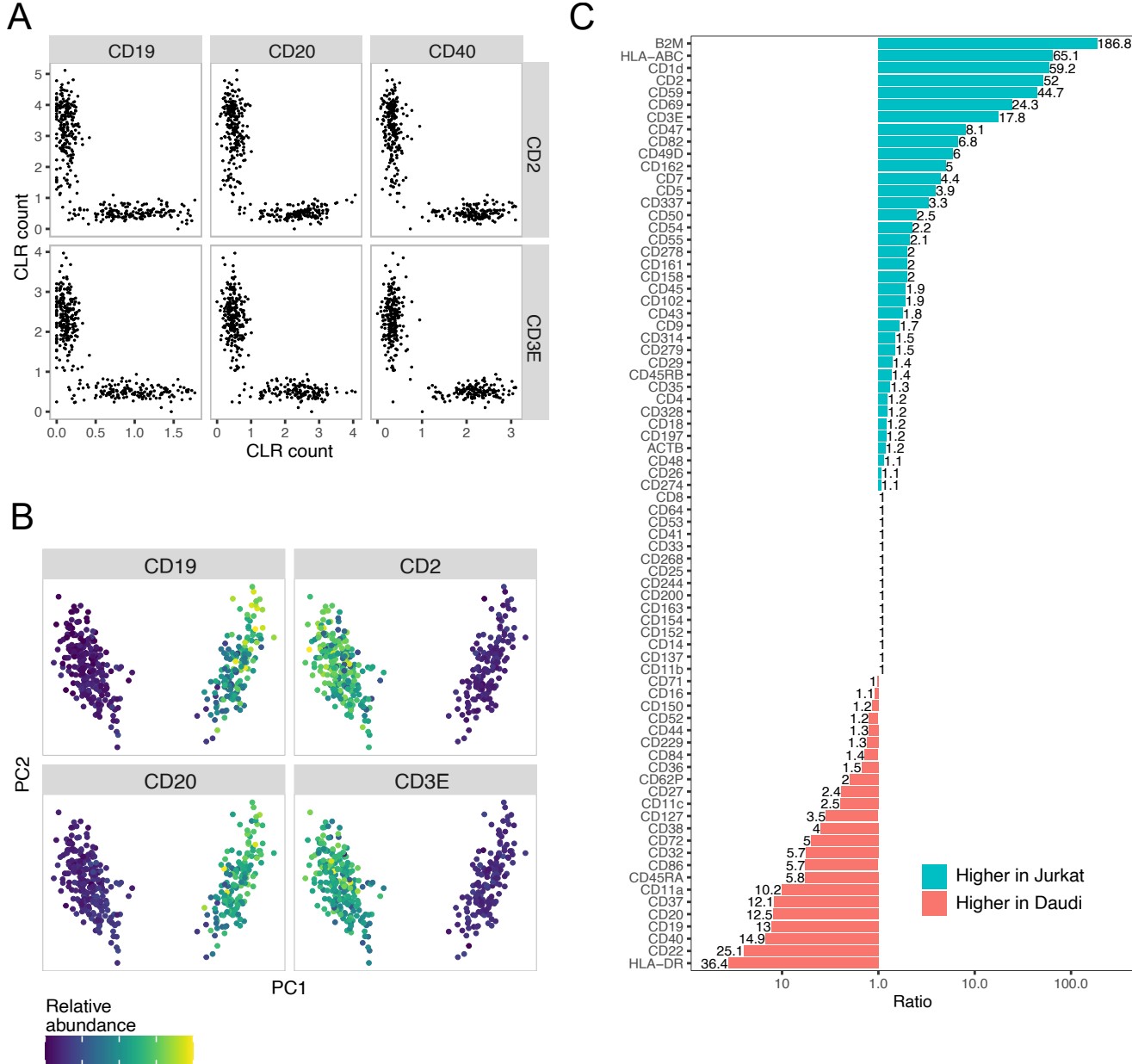

**Extended Data Fig. 2 | Doublets from MPX. a**) Scatter plots of the CLR counts of cells in a 50/50 Jurkat and Daudi cell line mixture, showing B cell markers along the x axis, and T cell markers along the y axis. **b**) PCA of cells in the Daudi-Jurkat mixture. The color of each panel encodes the CLR counts of B cell markers (CD19, CD20) and T cell markers (CD2, CD3E) on each cell. CLR values are scaled from minimum to maximum per figure panel. **c**) Bar plot of the ratio of median counts between Jurkat and Daudi.

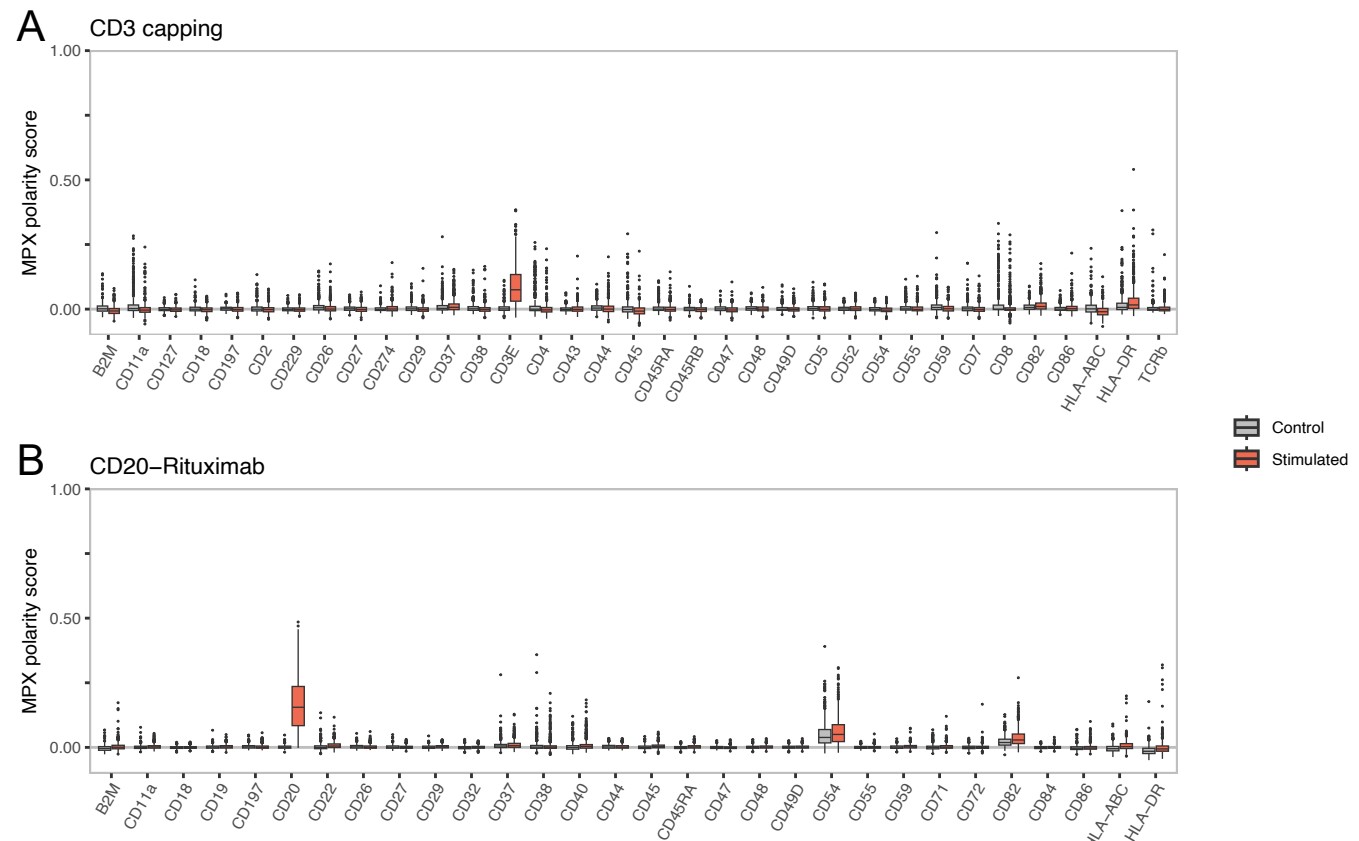

**Extended Data Fig. 3 | MPX Polarity Scores for stimulated and control conditions of all markers with total counts above isotype control levels.**
**a**) The CD3 capping experiment (619 unstimulated cells and 556 cells stimulated by CD3 capping; **b**) The Rituximab treatment experiment (440 unstimulated cells and 684 cells stimulated with Rituximab). Each box ranges the first and third quartiles (the 25th and 75th percentiles), with the median marked within the box. The whiskers range 1.5 IQRs from the box boundary, or to the outermost data point.

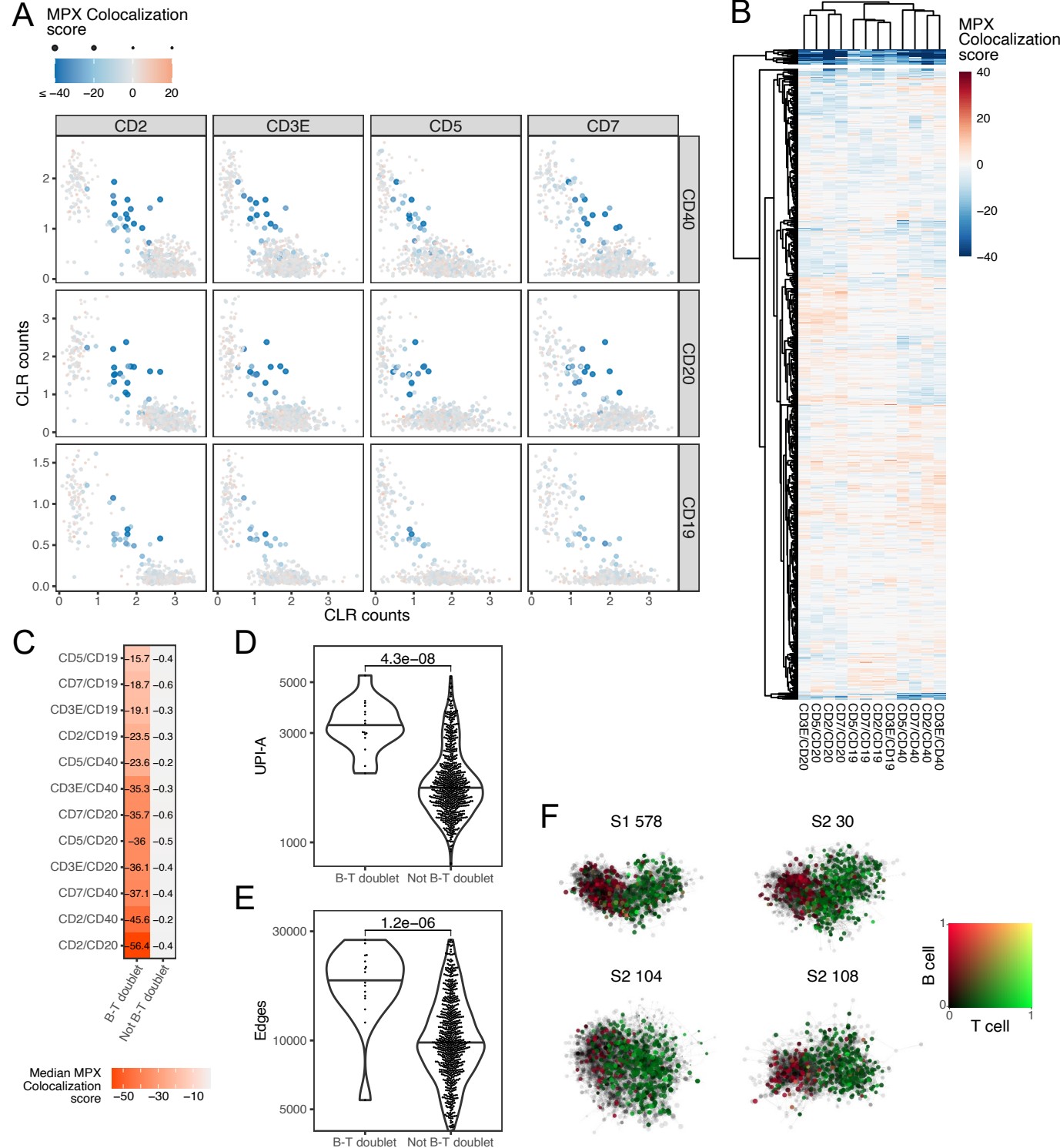

**Extended Data Fig. 4 | Analysis of B-T cell complexes using colocalization scores.** Cells have been gated to fulfill either CD3E > 0.8 or CD19 > 0.45 CLR counts to enrich for B and T cells. **a**) Scatterplots showing the abundance of B cell specific markers (CD19, CD20, CD40) and T cell specific markers (CD2, CD3E, CD5, CD7) within the PMBC dataset. A small subset of cells containing both B and T cell markers also exhibit low colocalization scores between the marker pairs, indicating spatial segregation within the component. **b**) Heatmap of the colocalization scores of all cells (capped to −40). Hierarchical clustering (agglomerative with complete linkage) identifies a cluster of 16 cells with strongly negative colocalization scores across all marker pairs. **c**) Heatmap showing the median MPX colocalization score. **d, e**) Violin plots of cell sizes in terms of number of protein molecules (edges) and UPI-As. Cells identified as B-T complexes were approximately twice as large as non-complexes in terms of both the number of edges (median 9757 and 18894, respectively; two-sided Wilcoxon Rank Sum test) and the number of UPI-A (median 1737 and 3166, respectively; two-sided Wilcoxon Rank Sum test). **f**) Graph visualization of 4 components identified as B-T complexes.

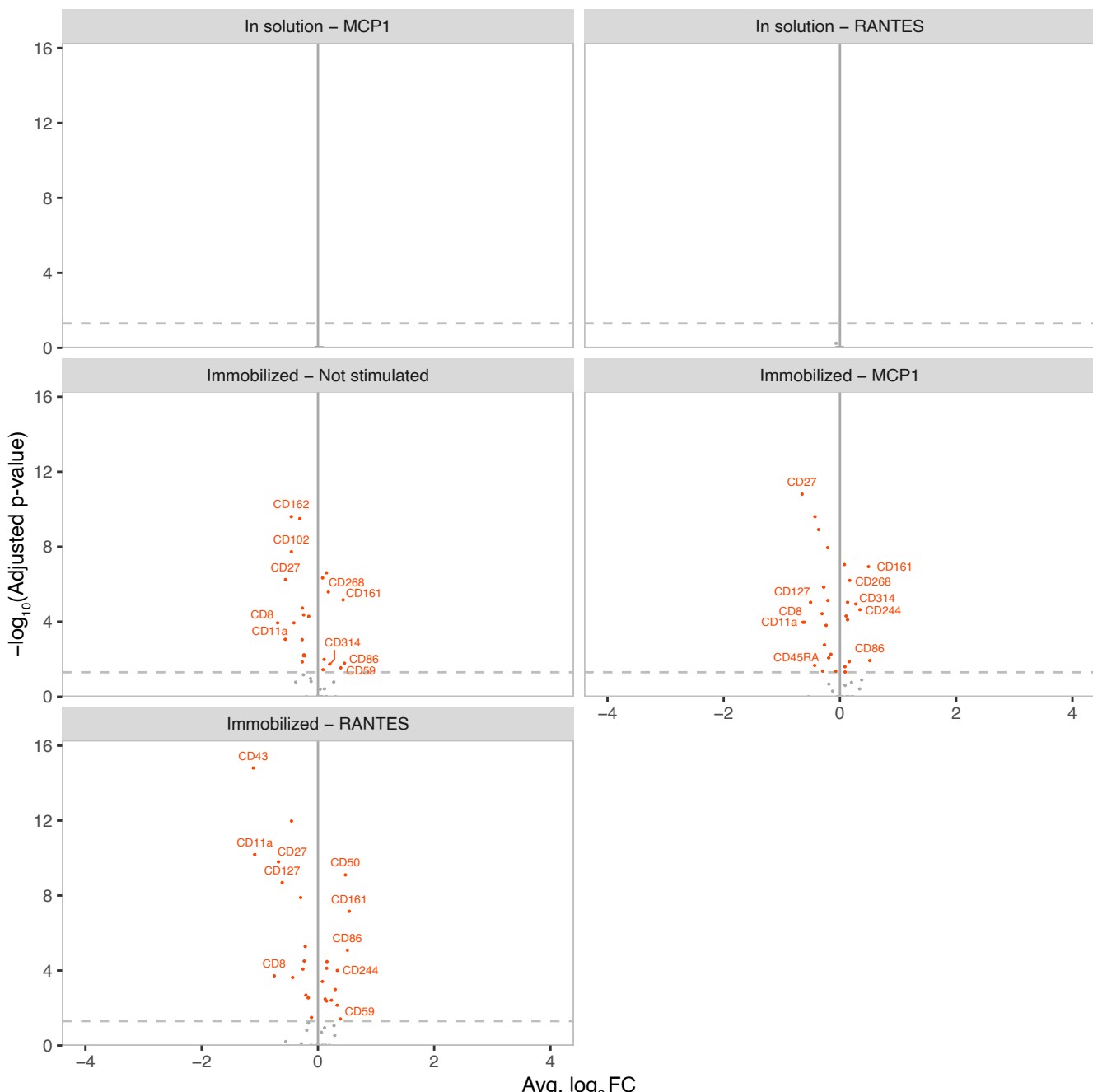

**Extended Data Fig. 5 | Volcano plots of uropod differential abundance analysis.** The analysis is performed per condition, each compared to the control 'In solution - Not stimulated'. The y-axis depicts Bonferroni adjusted p-value, and the x-axis represents the average log2 fold change. The 5 statistically significant (Bonferroni adjusted p-value < 0.05; two-sided Wilcoxon Rank Sum test) markers with highest effect size in each direction are annotated.

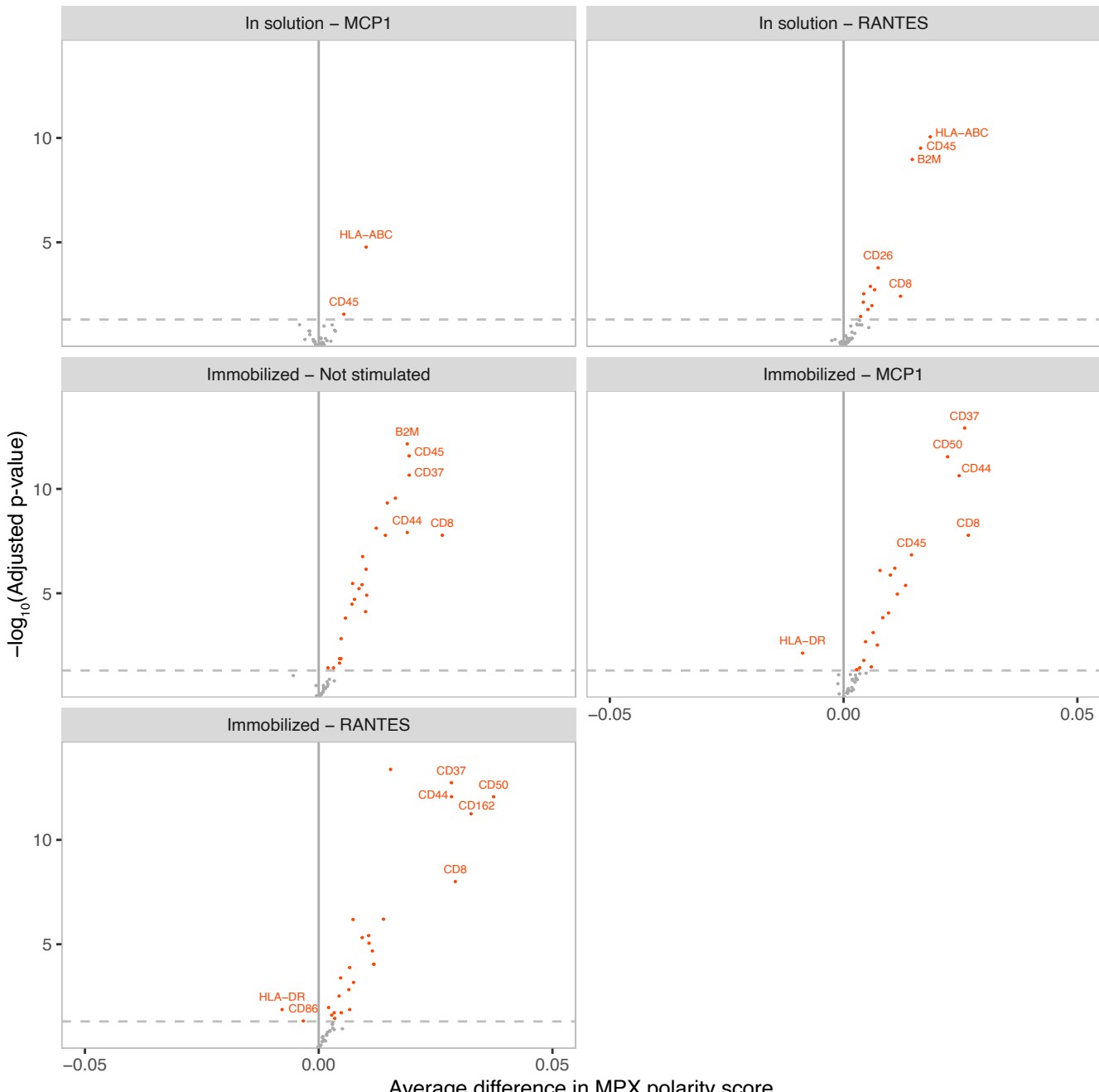

**Extended Data Fig. 6 | Volcano plots of uropod differential polarity analysis.** The analysis is performed per condition, each compared to the control 'In solution - Not stimulated'. The y-axis depicts Benjamini-Hochberg adjusted p-value, and the x-axis represents the average difference. The 5 statistically significant (BH adjusted p-value < 0.05; two-sided Wilcoxon Rank Sum test) markers with highest effect size in each direction are annotated.

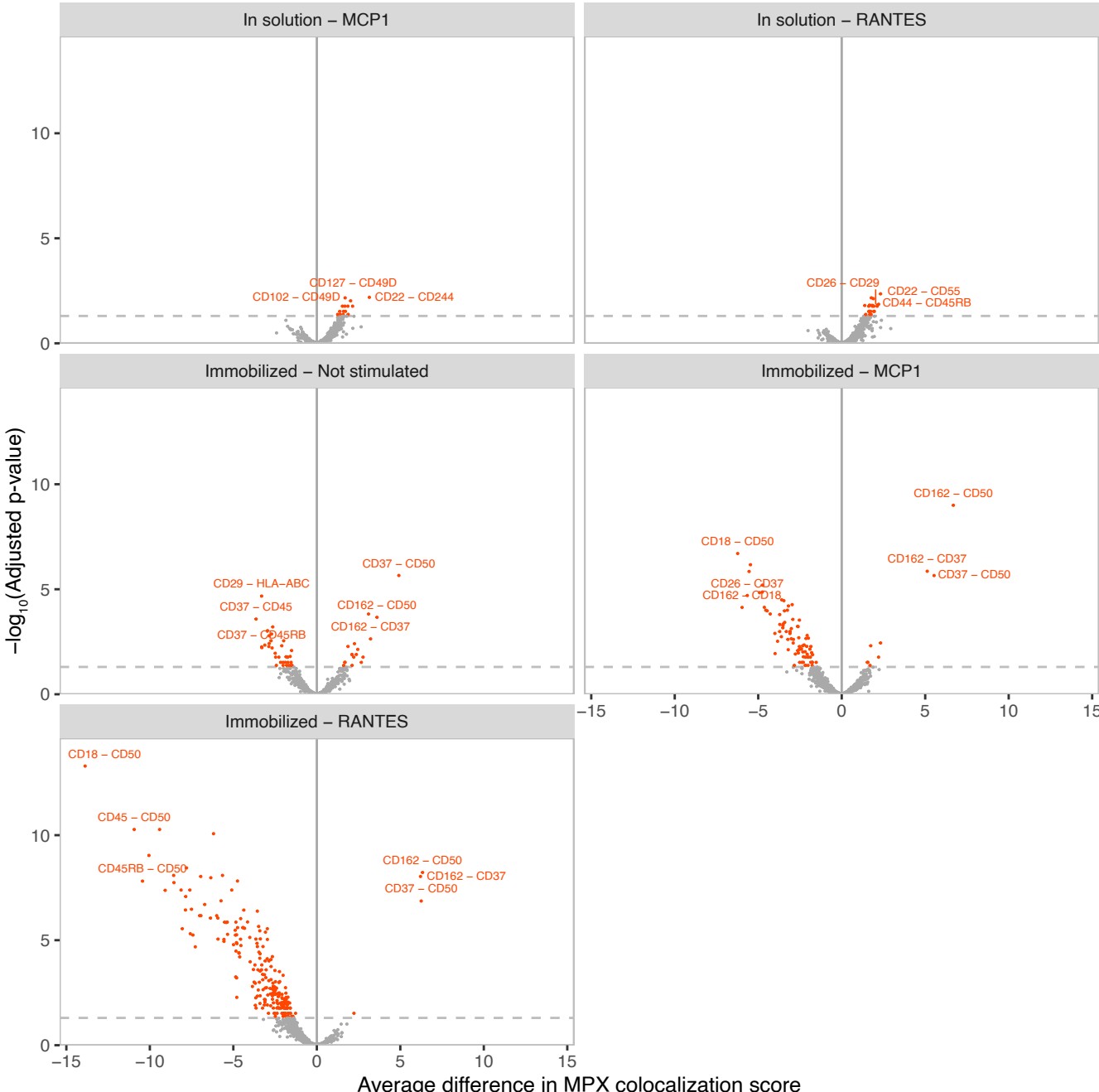

**Extended Data Fig. 7 | Volcano plots of uropod differential colocalization analysis.** The analysis is performed per condition, each compared to the control 'In solution - Not stimulated'. The y-axis depicts Benjamini-Hochberg adjusted p-value, and the x-axis represents the average difference. The 3 statistically significant (BH adjusted p-value < 0.05; two-sided Wilcoxon Rank Sum test) markers with highest effect size in each direction are annotated.

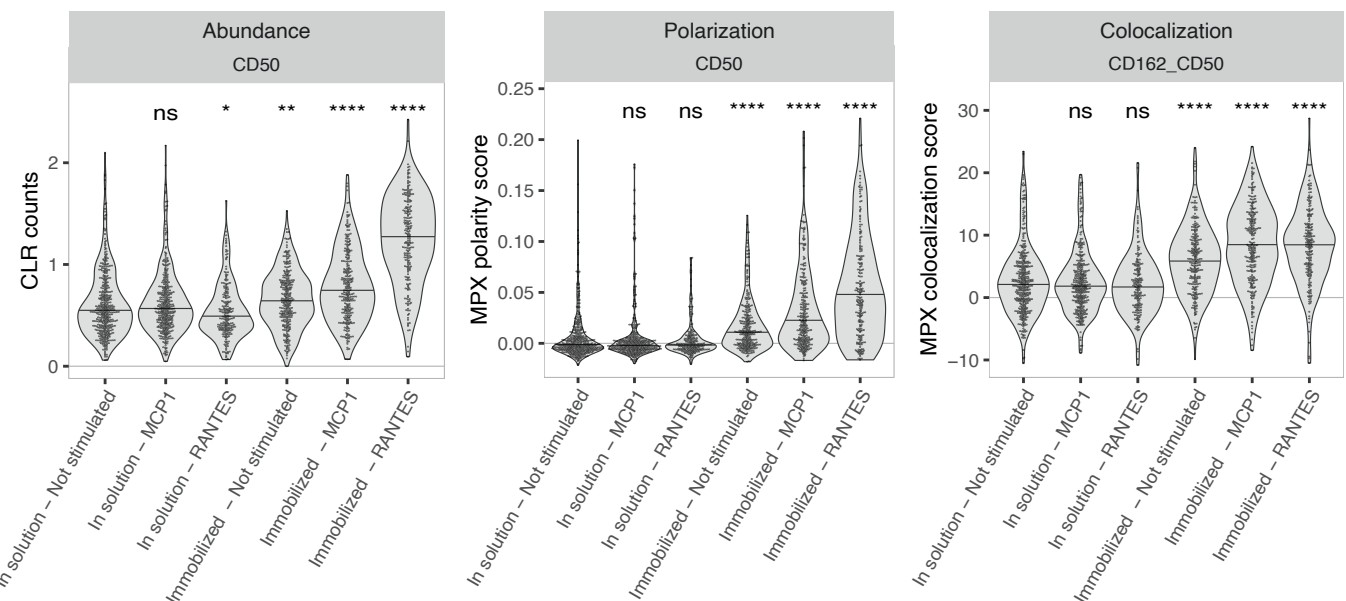

**Extended Data Fig. 8 | Violin plots of an example of a differentially abundant and polarized marker, and a differentially colocalized marker pair across conditions.** The level of the two-sided Wilcoxon Rank Sum test p-value is denoted as follows: 'ns' = not significant, '*' = p < 0.05, '**' = p < 0.01, '***' = p < 0.001, '****' = p < 0.0001. Respective p-values are (from left to right): Abundance: 0.3745, 0.0151, 0.0029, 2.2e-11, <2e-16; Polarization: 0.23, 0.59, <2e-16, <2e-16, <2e-16; Colocalization: 0.56, 0.34, 5.7e-15, <2e-16, <2e-16.

## Dispersed markers

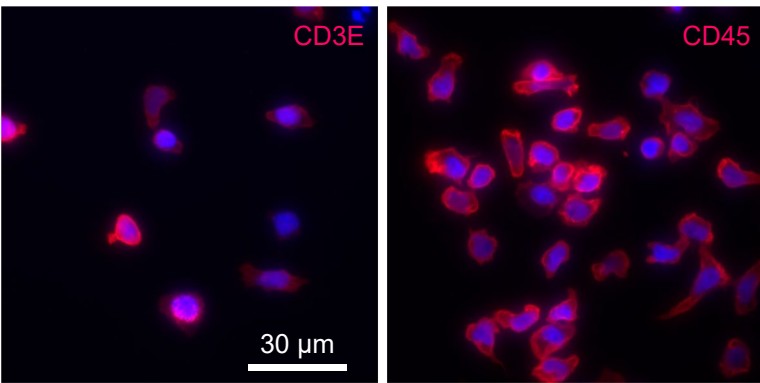

## Uropod markers

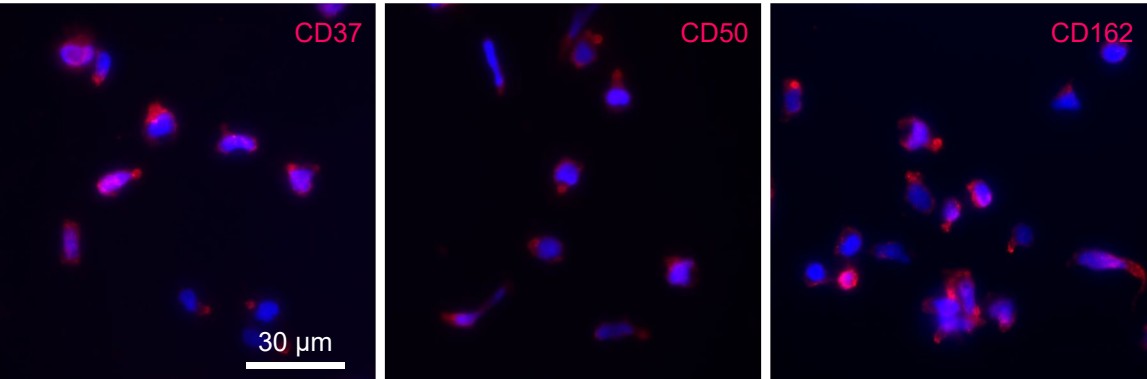

**Extended Data Fig. 9 | Fluorescence microscopy validation of MPX data on uropod formation in T cells immobilized on an ICAM1 coated surface and stimulated with RANTES.** The fixed cells were prepared as described for immobilized cells stimulated with RANTES, separately from two donors. The cells were then incubated with 5 µg/ml of the AOC for the indicated target for 40 minutes at 4 °C, followed by staining using a secondary antibody and imaging as described in Microscopy validation section of Methods. Top: Dispersed control markers CD3E and CD45 show dispersed localization on the cell surface. Bottom: Uropod markers CD37, CD50, and CD162 display spatial clustering toward the uropod bulge. Both donors displayed similar results.

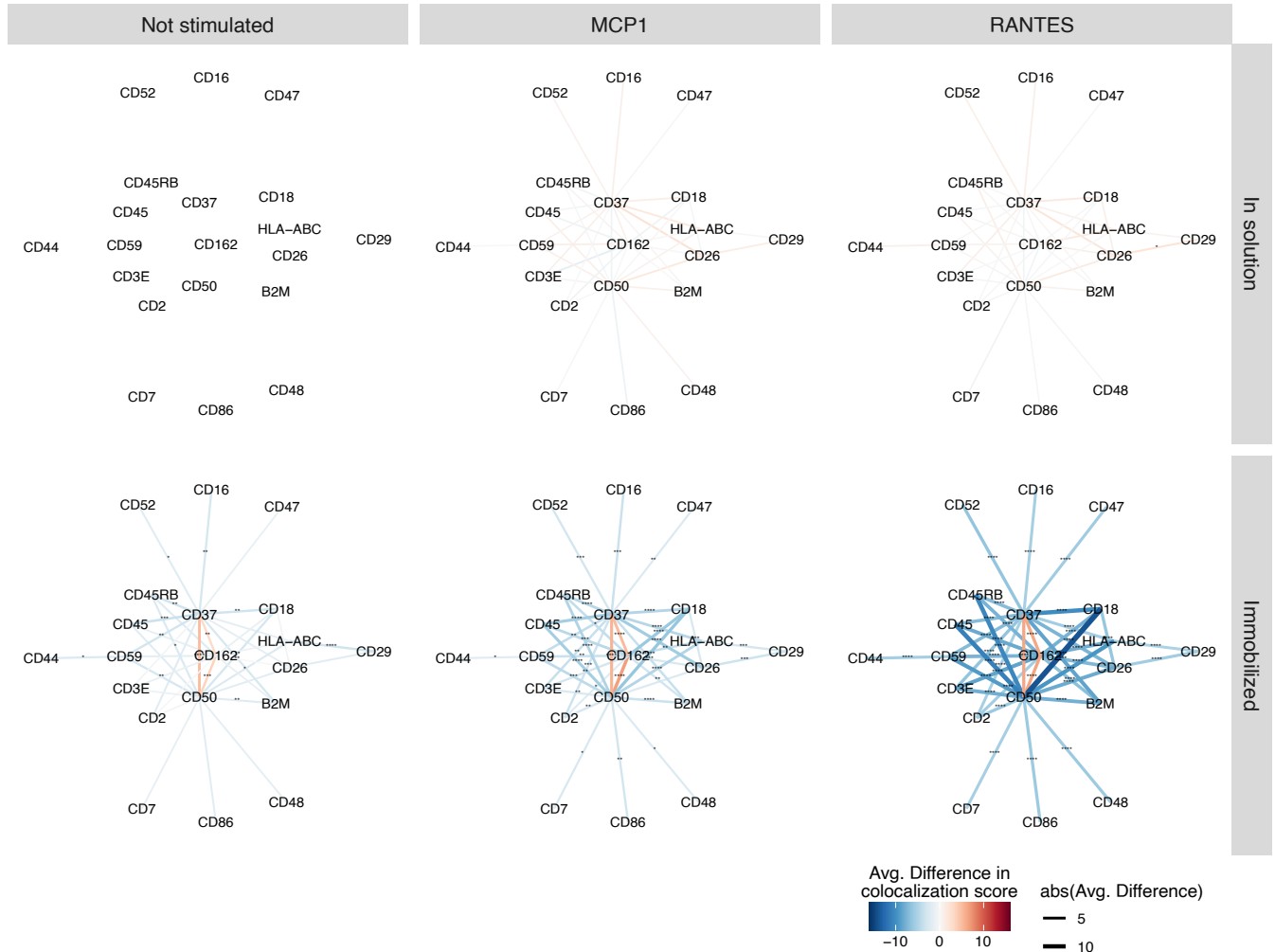

**Extended Data Fig. 10 | Network visualizations of differentially colocalized protein pairs.** Protein pairs that are differentially colocalized across any of the 5 conditions (CD54 coating and/or chemokine stimulation) compared to the in solution unstimulated control sample. The difference (increase or decrease) in colocalization between a pair of proteins is shown as the color of the link (blue = decrease, red = increase), and both the size of the link and its color intensity shows the magnitude of the difference. The level of the two-sided Wilcoxon Rank Sum test Benjamini Hochberg adjusted p-value is denoted as follows: '*' = p < 0.05, '**' = p < 0.01, '***' = p < 0.001, '****' = p < 0.0001 (See Supplementary Table 4 for exact p-values). Colocalization differences that are not statistically significant are still shown, but without the p-value annotation.

| | |
|---|---|

# Reporting Summary

## Statistics

For all statistical analyses, confirm that the following items are present in the figure legend, table legend, main text, or Methods section.

| n/a | Confirmed | |
|---|---|---|
| ☐ | ☒ | The exact sample size (*n*) for each experimental group/condition, given as a discrete number and unit of measurement |
| ☐ | ☒ | A statement on whether measurements were taken from distinct samples or whether the same sample was measured repeatedly |
| ☐ | ☒ | The statistical test(s) used AND whether they are one- or two-sided<br>*Only common tests should be described solely by name; describe more complex techniques in the Methods section.* |
| ☒ | ☐ | A description of all covariates tested |
| ☐ | ☒ | A description of any assumptions or corrections, such as tests of normality and adjustment for multiple comparisons |
| ☐ | ☒ | A full description of the statistical parameters including central tendency (e.g. means) or other basic estimates (e.g. regression coefficient) AND variation (e.g. standard deviation) or associated estimates of uncertainty (e.g. confidence intervals) |
| ☐ | ☒ | For null hypothesis testing, the test statistic (e.g. *F*, *t*, *r*) with confidence intervals, effect sizes, degrees of freedom and *P* value noted<br>*Give P values as exact values whenever suitable.* |
| ☒ | ☐ | For Bayesian analysis, information on the choice of priors and Markov chain Monte Carlo settings |
| ☒ | ☐ | For hierarchical and complex designs, identification of the appropriate level for tests and full reporting of outcomes |
| ☐ | ☒ | Estimates of effect sizes (e.g. Cohen's *d*, Pearson's *r*), indicating how they were calculated |

*Our web collection on statistics for biologists contains articles on many of the points above.*

## Software and code

Policy information about availability of computer code

| Data collection | All data was generated in-house. |
|---|---|
| Data analysis | FASTQ files were processed with an in-house software Pixelator (v0.12.0; https://github.com/PixelgenTechnologies/pixelator). Downstream analysis was performed in R (v4.2.2) using broom (v1.0.2), rstatix (v0.7.1), Seurat (v4.3.0), SeuratObject (v4.1.3), tidygraph (v1.2.2), and tidyverse (v1.3.2) for analysis, and ggforce (v0.4.1), ggplot2 (v3.4.0), ggplotify (v0.1.0), ggpubr (v0.5.0), ggraph (v2.1.0), graphlayouts (v0.8.4), igraph (v1.3.5), patchwork (v1.1.2), pheatmap (v1.0.12), plotly (v4.10.1), viridis (v0.6.2), and viridisLite (v0.4.1) for data visualization.<br><br>All analysis code can be found at https://github.com/PixelgenTechnologies/pixelgen-MPX-paper |

For manuscripts utilizing custom algorithms or software that are central to the research but not yet described in published literature, software must be made available to editors and reviewers. We strongly encourage code deposition in a community repository (e.g. GitHub). See the Nature Portfolio guidelines for submitting code & software for further information.

## Data

Policy information about availability of data

All manuscripts must include a data availability statement. This statement should provide the following information, where applicable:
- Accession codes, unique identifiers, or web links for publicly available datasets
- A description of any restrictions on data availability
- For clinical datasets or third party data, please ensure that the statement adheres to our policy

The MPX raw read data and Pixelator 0.12 processed output can be downloaded from Pixelgen Technologies (https://software.pixelgen.com/datasets/). Datasets are granted under a Creative Commons Attribution (https://creativecommons.org/licenses/by/4.0/legalcode) license.

## Human research participants

Policy information about studies involving human research participants and Sex and Gender in Research.

| | |
|---|---|
| Reporting on sex and gender | Data on neither sex nor gender were collected. |
| Population characteristics | no population studies were done |
| Recruitment | All PBMC samples were purchased from a Karolinska Hospital blood bank drawn from healthy volunteers with informed consent and withheld sample identity or other medical information. |
| Ethics oversight | *Identify the organization(s) that approved the study protocol.* |

Note that full information on the approval of the study protocol must also be provided in the manuscript.

# Field-specific reporting

Please select the one below that is the best fit for your research. If you are not sure, read the appropriate sections before making your selection.

☒ Life sciences    ☐ Behavioural & social sciences    ☐ Ecological, evolutionary & environmental sciences

For a reference copy of the document with all sections, see nature.com/documents/nr-reporting-summary-flat.pdf

# Life sciences study design

All studies must disclose on these points even when the disclosure is negative.

| | |
|---|---|
| Sample size | Since the study was primarily focused on method development, no calculations were carried out to determine the size of the biological samples. Previous experiments indicated that 500-1000 cells were sufficient to gain reproducible statistical significance in both abundance and spatial metrics for primary PBMC cell populations. For this methods development context, this range of cell input was therefore chosen. |
| Data exclusions | Cells were filtered to remove cells with few detected antibodies, and suspected antibody aggregates as outlined in Methods. |
| Replication | PBMC samples were processed in duplicates and cells were aggregated from both replicates for analysis. All replications were successful. |
| Randomization | Samples were randomly downsampled in silico for statistical tests. |
| Blinding | Blinding was not performed as this work is methods development. |

# Reporting for specific materials, systems and methods

We require information from authors about some types of materials, experimental systems and methods used in many studies. Here, indicate whether each material, system or method listed is relevant to your study. If you are not sure if a list item applies to your research, read the appropriate section before selecting a response.

## Materials & experimental systems

| n/a | Involved in the study |
|-----|----------------------|
| ☐ | ☒ Antibodies |
| ☐ | ☒ Eukaryotic cell lines |
| ☒ | ☐ Palaeontology and archaeology |
| ☒ | ☐ Animals and other organisms |
| ☒ | ☐ Clinical data |
| ☒ | ☐ Dual use research of concern |

## Methods

| n/a | Involved in the study |
|-----|----------------------|
| ☒ | ☐ ChIP-seq |
| ☒ | ☐ Flow cytometry |
| ☒ | ☐ MRI-based neuroimaging |

## Antibodies

| Antibodies used | All antibodies purchased from Antibodies-online and used at 5ug/ml)<br><br>B2M B2M-02<br>BAFFR 11C1<br>CD102 CBR-IC2-2<br>CD11a HI111<br>CD11b ICRF44<br>CD11c Bu15<br>CD127 A019D5<br>CD137 4B4-1<br>CD14 61D3<br>CD150 A12 (7D4)<br>CD152 BNI3<br>CD154 24-31<br>CD158 NKVFS1<br>CD16 3G8<br>CD161 HP-3G10<br>CD162 KPL-1<br>CD163 GHI-61<br>CD18 TS1-18<br>CD19 SJ25-C1<br>CD197 G043H7<br>CD1d 51-1<br>CD2 TS1-8<br>CD20 2H7<br>CD200 OX-104<br>CD22 S-HCL-1<br>CD229 HLy9-25<br>CD244 C1.7<br>CD25 BC96<br>CD26 BA5b<br>CD27 LT27<br>CD274 2A3<br>CD278 C398-4A<br>CD279 EH12-2H7<br>CD29 3B6<br>CD314 1D11<br>CD32 FUN2<br>CD33 WM53<br>CD337 P30-15<br>CD35 E11<br>CD36 5-271<br>CD37 IPO-24<br>CD38 HIT2<br>CD3E UCHT1<br>CD4 OKT4<br>CD40 C40-1605<br>CD41 A2A9-6<br>CD43 MEM-59<br>CD44 F10-44-2<br>CD45 HI30<br>CD45RA HI100<br>CD45RB MEM55<br>CD47 B6H12-2<br>CD48 MEM-102<br>CD49D 9F10<br>CD5 UCHT2<br>CD50 MEM-171<br>CD52 HI186 |
|---|---|

CD53 MEM-53
CD54 1H4
CD55 F4-29D9
CD59 MEM-43
CD62P AK4
CD64 10.1
CD69 FN50
CD7 4H9
CD71 CY1G4
CD72 3F3
CD8 SK1
CD82 C33
CD84 CD84-1-21
CD86 IT2-2
CD9 MEM-61
HLA-ABC W6-32
HLA-DR L243
SIGLEC S7-7
TCRb MEM-262
ACTB 137CT26-1-1
mIgG1 MOPC-21
mIgG2a MOPC-173
mIgG2b MPC-11

Validation    All antibodies are mouse monoclonal and  validated by the supplier antibodies-online.com for use in flow cytometry by their specific binding to respective target proteins found on a certain cell type in PBMC. Each antibody was revalidated by the authors in the same way on PFA fixed cells.

# Eukaryotic cell lines

Policy information about cell lines and Sex and Gender in Research

Cell line source(s)    Purchased (Raji, Jurkat, Daudi) from DSMZ (www.dsmz.de)

Authentication    All cell lines used (Raji, Jurkat, Daudi) have been verified by flow cytometry.

Mycoplasma contamination    All cell lines (Raji, Jurkat, Daudi) have tested negative for mycoplasma contamination prior to the experiments.

Commonly misidentified lines    None of the cell lines used (Raji, Jurkat, Daudi) are commonly misidentified according to the ICLAC register.
(See ICLAC register)

