## [Peer Review File · Nature Methods]

Peer Review Information

Manuscript Title: Molecular Pixelation: Spatial proteomics of single-cells by sequencing

Corresponding author name(s): Simon Fredriksson, Filip Karlsson

Editorial Notes: None

Reviewer Comments & Decisions:

Decision Letter, initial version:

Dear Simon,

Your Article entitled "Molecular Pixelation: Spatial proteomics of single-cells by sequencing" has now been seen by 3 reviewers, whose comments are attached. While they find your work of potential interest, they have raised serious concerns which in our view are sufficiently important that they preclude publication of the work in Nature Methods, at least in its present form.

As you will see, the reviewers raise serious concerns about the technical advance presented by this method due to a missing validation and benchmarking. It will be also be important to show that Molecular Pixelation is generalizable to samples beyond suspension cells.

Should further experimental data allow you to fully address these criticisms we would be willing to look at a revised manuscript (unless, of course, something similar has by then been accepted at Nature Methods or appeared elsewhere). This includes submission or publication of a portion of this work somewhere else. We hope you understand that until we have read the revised paper in its entirety we cannot promise that it will be sent back for peer-review.

If you are interested in revising this manuscript for submission to Nature Methods in the future, please contact me to discuss your appeal before making any revisions. Otherwise, we hope that you find the reviewers' comments helpful when preparing your paper for submission elsewhere.

Sincerely,
Madhura

Madhura Mukhopadhyay, PhD
Senior Editor
Nature Methods

Reviewers' Comments:

Reviewer #1:

Remarks to the Author:

Karlsson et al. developed the protein MPX method, which is an upgraded version of the proximity barcoding assay, using DNA-tagged antibodies. Authors describe their methods can be used to infer the relative location of each AOC molecule in single-cell. In addition, the authors demonstrated MPX to quantify the degree of spatial clustering or polarization through antibody experiments. Finally, abundance, polarity, and colocalization of the target protein were studied on immune cells which would be the indicator of cell type. MPX should be a very useful method to investigate the map of membrane proteins. In order to claim that MPX identifies relative locations of many proteins and 3D spatial resolution, however, some issues should be addressed before considering this manuscript published in Nature Methods.

1. The concept of Molecular Pixelation should be explained in a more detailed manner. The essence of this paper is the concept of their methods. However, it is extremely difficult to understand MPX without the knowledge of DNA microcopy and proximity barcoding assay. Without reading the papers on DNA microcopy and proximity barcoding assay, the readers should be able to understand the concept of MPX from Figure 1. How DNA pixel sets work is completely missing in this paper. The authors should explain the concept of MPX in a more detailed manner in Figure 1, alike ref 12 and 13.
2. The authors presented PBMC data in Figure 2. As a method paper, the authors need to validate the concept of MPX first as ref 12 and 13 did. Using a well-established system, for example, expressing both GFP and RFP-tagged proteins (and mixing two different types of cells), the authors should perform proof of concept experiments (or validation experiments) of MPX, which is missing in this paper.
3. The authors need to compare the fluorescence image with the image obtained by MPX. In ref 12, they presented the comparison in Fig. 4 and Fig 5. The authors can do similar work using the data in Fig. 3 D and E or using new data. The presentation of comparison between the fluorescence image and MPX image is important for validation of MPX.
4. In line 13 of page 2, the authors state that DNA-Pixels are smaller than 100 nm in diameter, which seems to be based on EM images in Figure S14. The size of RCA produced ssDNA for DNA-Pixel on cell surface may be different from EM images in a vacuum. The authors need to give the resolution of DNA-Pixel by measuring directly or indirectly the size of DNA-Pixel on cell surface. It would be good to provide a comparison of the resolution between MPX and optical microscopy.
5. The strength of MPX is observing the interactions or colocalization of proteins. In Fig. 4D, the authors presented 3D animation. What are the average (spatial) distances between proteins, such as CD162, CD37, and CD50?

Reviewer #2:

Remarks to the Author:

Evaluation of Karlsson et al:

In this manuscript, Karlsson et al. present Molecular Pixelation (MPX), an optics-free DNA sequencing based approach for localizing antibodies against protein targets, on single-cells. Such an approach,

using antibody oligonucleotide conjugates (AOCs) and unique DNA molecular pixel barcodes, akin to DNA microscopy, is an interesting approach for subcellular “imaging” of proteins on individual cells. While it is no small feat, this reviewer finds some of the analysis to need more clarification, the lack of sufficient comparisons against existing approaches (e.g. CyTOF, CITE-seq, Olink/SPARC etc) or “ground truths”, clarification of the Material & Methods section, and most importantly, a clearer biological application/demonstration of MPX. Of note, there is an untapped potential of MPX in localizing antibodies beyond just within each individual cells, i.e. on a large scale, including in tissues. Major concerns:

1. The paper focuses on PBMCs or cell lines. It would seem MPX would have untapped potential beyond suspension cells, or actual dissociated tissues/diseased cells
2. Given the limited numbers of the cells analyzed with MPX (~500s per replicate), can the authors clarify the time/resources needed for MPX vs existing methods, and scalability of the method?
3. It is unclear from the paragraph starting on Line 26 Page 4, how the cell annotation was performed using the “differential abundance analysis” with downsampling is an uncommon method for single-cell protein-based annotation, and the figures showing average log₂FC could be better represented as z-scores.
 - a. Common methods including graph-based clustering, e.g. phenograph or flowSOM
 - b. Is there a reason for downsampling, given the already sparse total number of cells collected?
4. The authors should demonstrate a ground truth comparison, such as flow cytometry, CyTOF, CITE-seq or others to show consistencies in the PBMC compositions detected. This is also a good opportunity to compare signal-noise ratios or dynamic ranges
5. The ability to infer spatial localization through the graph based MPX data is remarkable, and while the authors showed some confirmations through microscopy, it is a missed opportunity to analyze the extent and distribution of the clustering from microscopy vs MPX over larger numbers and examples. Such metrics and comparisons will be key to highlight the uniqueness of MPX
6. Given most of the analysis in the paper are performed on spheroid cells (B/T cells etc), and the spheroid representation of the data, how does MPX perform on non spherical or abnormally shaped cells?

Minor concerns:

1. Much of the Abs are focused on surface proteins, it is not immediately clear to a reader whether/why, and the technical barriers towards that
2. It seems a missed opportunity to obtain RNA from the same cells, given the unique nature of MPX barcoding
3. Aspects of the analysis in the M&M are lacking, including
 - a. how antibodies were validated and titrated (only mentions one-by-one on PFC fixed PBMC, pre-conjugation). The DBCO-Azide click chemistry may well affect the binding efficiencies/capability of the antibodies, and no validations were performed after conjugations
 - b. Why were the 10 largest cells removed from each replicate? Is this empirically determined or a rough ballpark?
 - c. A “manual cutoff” was performed on the minimum number of detected UMIs. It is unclear if the subsequent description for “minimum component size cutoffs” is the same? The variation appears to be rather high across different cell types and experiments, which does not inspire confidence in large potential batch effects of MPX
 - d. In line 42 of Page 18, the clustering performed here is using the default method in Seurat, and appears to be different from the description made in the paragraph for line 26 of Page 4. Would the authors be able to help clarify and elaborate on why the differences?

Reviewer #3:
Remarks to the Author:
Karlsson et al.

Karlsson et al. introduces a novel method called Molecular Pixelation (MPX) for ascertaining the relative location of proteins on cell membranes. MPX utilizes Antibody Oligonucleotide Conjugates (AOCs), DNA-tagged antibodies that selectively bind to specific proteins. Through a series of steps involving the incorporation of unique molecular identifiers (UMIs) within DNA pixels, the technique enables spatial analysis of protein arrangement. The novelty of this approach lies in the formation of DNA pixels, each containing a Unique Pixel Identifier (UPI), allowing multiple DNA pixels to hybridize with neighboring AOCs. The UPI sequences are incorporated onto AOCs, forming neighborhoods. PCR amplification and sequencing of AOCs, containing a UMI and two UPI barcodes, indicate neighborhood membership. The relative location of unique AOCs is inferred from the overlap of UPI neighborhoods, and the data can be represented as a bipartite graph.

In our view, the developed MPX technique represents a significant breakthrough in the identification of intracellular protein locations, with the potential to yield novel biological insights. However, it is essential to acknowledge that the authors have, thus far, offered only limited validation of their method, primarily demonstrating increased autocorrelation in response to a stimulus *in vitro*. To make this manuscript a complete methods paper for a new technique, it is necessary for the authors to extend their validation. This should encompass a comprehensive examination of target performance as the complexity of the panel increases, an exploration of spatial resolution, and an evaluation of the technique's sensitivity to variations in target abundance.

Major Comments:

- All the major concerns for this work are regarding the validation. In brief, Figure 3 in its current form is insufficient validation to suggest that the techniques work.

Concerns on Validation / Limitations:

1. Presumably the antibodies in this assay could be targeting proteins that differ in abundance by several orders of magnitude. How do these disparities affect assay performance? Experimental data demonstrating how this assay performs with large differences in protein concentration are needed.
2. How difficult is it to optimize this assay? How sensitive is assay performance to antibody titers? Most cytometric approaches involving surface staining of antibodies require multiple rounds of titration, is this true here? Experiments where the antibody staining concentrations are varied independent of one another within a reasonable range (i.e. 0.25x – 4x) to characterize how this impacts what is detected is needed.
3. What is the sequencing cost? Can the authors provide some general heuristics about the logistics around this?
4. It doesn't appear that there is any validation in the manuscript to determine how close two proteins need to be to be detected as neighbors. Quantitative experiments to determine the radius of interaction for this process is a critical part of this paper and should be added.
5. How is assay performance impacted by the number of antibodies used for staining? Replicate

experiments with smaller subsets of reagents should be performed to quantitatively determine how this impacts assay performance.

Author Rebuttal to Initial comments

December 1, 2023

Karlsson *et al.* Molecular Pixelation: Spatial proteomics of single-cells by sequencing

Dear Editor and Reviewers,

We greatly appreciate all the comments, suggestions, and encouraging remarks about our manuscript. To answer the comments point-by-point, our edits in the revised manuscript are described below in red and shown in the resubmitted manuscript as red text

A few new Supplementary figures and videos have been added to improve understanding of MPX and its performance. A suggested front page cover art is also included.

Best regards,
Simon Fredriksson

Reviewers' Comments:

Reviewer #1:

Remarks to the Author:

Karlsson *et al.* developed the protein MPX method, which is an upgraded version of the proximity barcoding assay, using DNA-tagged antibodies. Authors describe their methods can be used to infer the relative location of each AOC molecule in single-cell. In addition, the authors demonstrated MPX to quantify the degree of spatial clustering or polarization through antibody experiments. Finally, abundance, polarity, and colocalization of the target protein were studied on immune cells which would be the indicator of cell type. MPX should be a very useful method to investigate the map of membrane proteins. In order to claim that MPX identifies relative locations of many proteins and 3D spatial resolution, however, some issues should be addressed before considering this manuscript published in Nature Methods.

1. The concept of Molecular Pixelation should be explained in a more detailed manner. The essence of this paper is the concept of their methods. However, it is extremely difficult to understand MPX without the knowledge of DNA microcopy and proximity barcoding assay. Without reading the papers on DNA microcopy and proximity barcoding assay, the readers should be able to understand the concept of MPX from Figure 1. How DNA pixel sets work is completely missing in this paper. The authors should explain the concept of MPX in a more detailed manner in Figure 1, alike ref 12 and 13.

We thank the reviewer for this feedback. We agree that the manuscript can benefit from a deeper description and more details and we have thoroughly revised Figure 1 and the manuscript text. The new figure 1A includes a description of the assay in molecular detail as well as at a "macro" level. We have also included a figure 1B, which explains the relationship between the resulting MPX cell graph and the individual AOCs.

2. The authors presented PBMC data in Figure 2. As a method paper, the authors need to validate the concept of MPX first as ref 12 and 13 did. Using a well-established system, for example, expressing both GFP and RFP-tagged proteins (and mixing two different types of cells), the authors should perform proof of concept experiments (or validation experiments) of MPX, which is missing in this paper.

Similar to the GFP/RFP experiments suggested by Reviewer #1, we did validate MPX by mixing two different cell lines (B-cell Daudi and T-cell Jurkat) in Supplementary Figure S5 to show the duplet rate. As a side product of that experiment, no cells were incorrectly defined and the scatter plot of incompatible B vs T:cell markers highlight the specificity of MPX and the AOC reagents. This has now been clarified in the new text on page 5. The suggested GFP and RFP tagged cell mixing would yield the same type of validation results as the presented B-cell : T-cell line mixing experiment.

To validate the spatial performance of MPX we conducted the well known CD3 capping experiment and CD20 polarization Rituximab experiment which behaved as according to the literature and our side-by-side validation by microscopy shown in Figure 3.

3. The authors need to compare the fluorescence image with the image obtained by MPX. In ref 12, they presented the comparison in Fig. 4 and Fig 5. The authors can do similar work using the data in Fig. 3 D and E or using new data. The presentation of comparison between the fluorescence image and MPX image is important for validation of MPX.

We have included five additional examples corresponding to the same markers as highlighted in the uropod experiment in figure 4D. Immunofluorescence images of T cells stimulated with ICAM1 immobilization and RANTES of CD3E, CD45, CD37, CD50, and CD162 are now provided in Supplementary Figure S14, which confirm localization of CD37, CD50, and CD162 to the uropod, while CD3E and CD45 are dispersed.

4. In line 13 of page 2, the authors state that DNA-Pixels are smaller than 100 nm in diameter, which seems to be based on EM images in Figure S14. The size of RCA produced ssDNA for DNA-Pixel on cell surface may be different from EM images in a vacuum. The authors need to give the resolution of DNA-Pixel by measuring directly or indirectly the size of DNA-Pixel on cell surface. It would be good to provide a comparison of the resolution between MPX and optical microscopy.

This is a very good point. Because these DNA-based Pixels are polymeric molecules and not of solid state of an absolutely defined size, it is not as straightforward to measure the resolution as we would like. We have attempted to measure the size of a DNA-pixel bound to a cell using SEM. However, the background noise of the cellular biomaterial prevented us from seeing the DNA-pixel. However, they are detectable when placed on a planar surface.

We have added a section on the resolution of MPX in Supplementary Figure S17.

5. The strength of MPX is observing the interactions or colocalization of proteins. In Fig. 4D, the authors presented 3D animation. What are the average (spatial) distances between proteins, such as CD162, CD37, and CD50?

This is an interesting question. We like to point out that we do not claim to detect interactions in our manuscript, but dynamic changes in colocalization. From our data type we are currently not able to provide a precise average distance in nm but we provide degrees of separation in the graph-data. As our method is chemical (or molecular) we can only provide an estimate of resolution, Supplementary Figure S17.

Reviewer #2:

Remarks to the Author:

Evaluation of Karlsson et al:

In this manuscript, Karlsson et al. present Molecular Pixelation (MPX), an optics-free DNA sequencing based approach for localizing antibodies against protein targets, on single-cells. Such an approach, using antibody oligonucleotide conjugates (AOCs) and unique DNA molecular pixel barcodes, akin to DNA microscopy, is an interesting approach for subcellular "imaging" of proteins on individual cells.

While it is no small feat, this reviewer finds some of the analysis to need more clarification, the lack of sufficient comparisons against existing approaches (e.g. CyTOF, CITE-seq, Olink/SPARC etc) or "ground truths", clarification of the Material & Methods section, and most importantly, a clearer biological application/demonstration of MPX. Of note, there is an untapped potential of MPX in localizing antibodies beyond just within each individual cells, i.e. on a large scale, including in tissues.

Major concerns:

1. The paper focuses on PBMCs or cell lines. It would seem MPX would have untapped potential beyond suspension cells, or actual dissociated tissues/diseased cells

We agree, there is much untapped potential here.

Ongoing work in our laboratory has shown feasibility to use MPX for RNA detection in single cells, and proteins in FFPE tissues. We have added to the Summary section on page 14 that "FFPE tissue is (in progress)". The data we have to date is too premature for inclusion in the current manuscript.

We have also added data on B:T-cell complexes found in PBMC providing an additional application of MPX beyond single cells, and which also highlights that spatial metrics can improve the detection of cell complexes over what just abundance can do. This is found in Supplementary Figure S9

Diseased cells in the form of cancer cell lines from blood cancers have been analyzed and data is present in the manuscript.

2. Given the limited numbers of the cells analyzed with MPX (~500s per replicate), can the authors clarify the time/resources needed for MPX vs existing methods, and scalability of the method?

We have added a statement in the Summary section on page 14 regarding how many sequencing reads are required by each cell, which is 120,000 reads, about 2-3 fold more than a typical RNAseq experiment. The protocol takes 2 days in the lab and the method is automatable and scalable. More cells can be analyzed by more sequencing, placing throughput further beyond what is possible with a microscope.

3. It is unclear from the paragraph starting on Line 26 Page 4, how the cell annotation was performed. using the "differential abundance analysis" with downsampling is an uncommon method for single-cell protein-based annotation, and the figures showing average log2FC could be better represented as z-scores.

We have now exchanged heatmaps showing the average log2FC to Z-scores, and indeed the visualization is improved in interpretability. We thank the reviewer for this suggestion.

a. Common methods including graph-based clustering, e.g. phenograph or flowSOM

We thank the reviewer for this comment, and we agree that it is indeed unclear and we will rephrase the explanation accordingly. We do in fact perform graph-based clustering (Louvain) as the reviewer suggests, albeit this is not clearly stated. The differential abundance analysis is performed across cell populations as defined by the Louvain clustering to obtain proteins with statistically significant differences across cell populations. The differentially abundant markers have in turn been used for manual cell annotation. We have clarified this in the revised manuscript on page 4.

b. Is there a reason for downsampling, given the already sparse total number of cells collected?

The downsampling is performed for two reasons. First, to manage p-value inflation due to the large number of cells, and secondly to ensure that the number of cells are equal across compared conditions in statistical analyses.

Downsampling of cells has been performed using Seurat's own "max.cells.per.ident" parameter for differential abundance analysis, which is a default parameter that is often used for the reasons stated above, as well as to increase the speed of analysis in larger experiments with many variables.

4. The authors should demonstrate a ground truth comparison, such as flow cytometry, CyTOF, CITE-seq or others to show consistencies in the PBMC compositions detected. This is also a good opportunity to compare signal-noise ratios or dynamic ranges

Cell type compositions found in PBMC using MPX vs. Flow Cytometry has been added as Supplementary Figure S3. S/N and dynamic range comparisons are also included in Supplementary Table S2.

5. The ability to infer spatial localization through the graph based MPX data is remarkable, and while the authors showed some confirmations through microscopy, it is a missed opportunity to

analyze the extent and distribution of the clustering from microscopy vs MPX over larger numbers and examples. Such metrics and comparisons will be key to highlight the uniqueness of MPX

We thank the reviewer for pointing out this opportunity. We have included five additional examples corresponding to the same markers as highlighted in the uropod experiment in figure 4D. Immunofluorescence images of T cells stimulated with ICAM1 immobilization and RANTES of CD3E, CD45, CD37, CD50, and CD162 are now provided in Supplementary Figure S14 and commented on page 10. These confirm localization of CD37, CD50, and CD162 to the uropod, while CD3E and CD45 are dispersed.

6. Given most of the analysis in the paper are performed on spheroid cells (B/T cells etc), and the spheroid representation of the data, how does MPX perform on non spherical or abnormally shaped cells?

Data is projected into spheres for visualization purposes only. But spatially analyzed for Polarity and Colocalization as a graph without any assumptions of the cell shape. This has been clarified on page 19.

We are now presenting B-T cell complexes found in PBMC, which are not spherical, which is shown in the graph based data. This is described on page 9 and Supplementary Figure S9.

We also show a raw graph representation of control and uropod formed cells in Supplementary Video 1 and 2.

Minor concerns:

1. Much of the Abs are focused on surface proteins, it is not immediately clear to a reader whether/why, and the technical barriers towards that

We chose to optimize MPX on- and simultaneously apply it to the immune cell surface proteins as they are understudied from the spatial perspective and good antibodies are available. Also, the dynamical distribution of surface proteins are well described and related to cellular activation, motility, signaling etc.

2. It seems a missed opportunity to obtain RNA from the same cells, given the unique nature of MPX barcoding

Very true, we have not prioritized such an experiment at this stage. Single cells could be captured and analyzed for RNA also but we find it outside of the scope of this work where we have focused on illustrating the technical performance of MPX and how the method can be applied in research. We are exploring the addition of mRNA as a possible added layer of data in the future although one can question whether the spatial distribution of mRNA molecules within a cell is as variable as that of proteins.

3. Aspects of the analysis in the M&M are lacking, including

a. how antibodies were validated and titrated (only mentions one-by-one on PFC fixed PBMC, pre-conjugation). The DBCO-Azide click chemistry may well affect the binding efficiencies/capability of the antibodies, and no validations were performed after conjugations

We thank the reviewer for this comment. We improved Materials and Methods page 17, by including more details on the validation. All antibodies were validated under the same conditions including concentration. So there is likely room for further improvement in S/N by individual titrations. When working with highly multiplexed assays, it is easier to use all AOC at the same concentration and if a particular antibody clone does not perform well at that concentration, it is simply replaced.

The antibodies used had well documented specificity prior to us selecting them. Validation after conjugations was performed using our MPX assay, ensuring that each individual marker showed a positive signal on a cell type where the marker is known to be expressed, and also confirming that no signal was seen on cell types where the marker should not be expressed.

b. Why were the 10 largest cells removed from each replicate? Is this empirically determined or a rough ballpark?

These cells were indeed removed as "a rough ballpark" that approximately corresponded to the higher tail in the cell size distribution. This type of filtering is a standard process for Flow Cytometry and single cell analysis as well. A description is now found on page 19.

c. A "manual cutoff" was performed on the minimum number of detected UMIs. It is unclear if the subsequent description for "minimum component size cutoffs" is the same? The variation appears to be rather high across different cell types and experiments, which does not inspire confidence in large potential batch effects of MPX

We thank the reviewer for this comment; we will clarify the phrasing here. The manual cutoff does indeed refer to the minimum component size cutoff. Pixelator finds a rough cutoff automatically as described in the section "Data analysis by Pixelator", using a similar methodology to what is routinely done in droplet based scRNAseq (analogous to Cellranger's Barcode Rank Plot: <https://support.10xgenomics.com/single-cell-gene-expression/software/pipelines/latest/advanced/barcode-rank-plot>) :

"From the total number of edges of each component membership, size outliers were identified based on the descending rank order distribution. A size threshold based on the rate of change was defined by finding the first and second derivatives from an univariate smooth spline curve fitted to the linear-log distribution of the ranked antibody count data. Edges for components

considered as size outliers were removed from the edgelist. The remaining components after size filtering were those considered to represent cells."

The manual cutoff refers to a refinement of that cutoff. Regarding the variance in cutoff, we observe differences in the number of UMIs and UPIs (pixels) depending on the studied cell type and cell state, which is largely explained by the cell surface size and protein density. Activated cells, and large cell types generally have a higher number of protein molecules detected on the cell surface. In our experiments, we see a lower number of protein molecules in unstimulated cells (4,000 for the unstimulated PBMC experiment, 5,000 for the CD3 capping experiment), while the cutoff is larger for activated cells (8,000 for the uropod experiment), and largest for the experiment with the largest cells (20,000 for the Raji experiment).

d. In line 42 of Page 18, the clustering performed here is using the default method in Seurat, and appears to be different from the description made in the paragraph for line 26 of Page 4. Would the authors be able to help clarify and elaborate on why the differences?

We thank the reviewer for pointing this out. We are indeed using the default method of graph based clustering in Seurat and the Materials and Methods (Page 17-18) is correct. We have further clarified what was done in the results part as suggested (Page 4).

Reviewer #3:

Remarks to the Author:

Karlsson et al.

Karlsson et al. introduces a novel method called Molecular Pixelation (MPX) for ascertaining the relative location of proteins on cell membranes. MPX utilizes Antibody Oligonucleotide Conjugates (AOCs), DNA-tagged antibodies that selectively bind to specific proteins. Through a series of steps involving the incorporation of unique molecular identifiers (UMIs) within DNA pixels, the technique enables spatial analysis of protein arrangement. The novelty of this approach lies in the formation of DNA pixels, each containing a Unique Pixel Identifier (UPI), allowing multiple DNA pixels to hybridize with neighboring AOCs. The UPI sequences are incorporated onto AOCs, forming neighborhoods. PCR amplification and sequencing of AOCs, containing a UMI and two UPI barcodes, indicate neighborhood membership. The relative location of unique AOCs is inferred from the overlap of UPI neighborhoods, and the data can be represented as a bipartite graph.

In our view, the developed MPX technique represents a significant breakthrough in the identification of intracellular protein locations, with the potential to yield novel biological insights. However, it is essential to acknowledge that the authors have, thus far, offered only limited validation of their method, primarily demonstrating increased autocorrelation in response to a stimulus *in vitro*. To make this manuscript a complete methods paper for a new technique, it is necessary for the authors to extend their validation. This should encompass a comprehensive examination of target performance as the complexity of the panel increases, an exploration of spatial resolution, and an evaluation of the technique's sensitivity to variations in target abundance.

We thank the reviewer for the kind words. We have now provided much further technical validation of MPX in the manuscript. See above

Major Comments:

- All the major concerns for this work are regarding the validation. In brief, Figure 3 in its current form is insufficient validation to suggest that the techniques work.

Concerns on Validation / Limitations:

1. Presumably the antibodies in this assay could be targeting proteins that differ in abundance by several orders of magnitude. How do these disparities affect assay performance? Experimental data demonstrating how this assay performs with large differences in protein concentration are needed.

For the PBMC dataset (Figure 2), we show that the assay works for a variety of cell types that differ in abundance by several orders of magnitude. Furthermore, we show the ability to detect differential spatial polarization in both CD3-capped T cells and Rituximab-treated Raji B cells

where the abundance levels differ. In Supplementary Figure S6C which shows results from Daudi and Jurkat cell lines, we have now also added ratios of count levels between Daudi and Jurkat, where some proteins differ >100 fold in count levels. Together, we believe these results illustrate that assay performance is robust to disparities in abundance levels.

2. How difficult is it to optimize this assay? How sensitive is assay performance to antibody titers? Most cytometric approaches involving surface staining of antibodies require multiple rounds of titration, is this true here? Experiments where the antibody staining concentrations are varied independent of one another within a reasonable range (i.e. 0.25x – 4x) to characterize how this impacts what is detected is needed.

The reviewer correctly points out that the method would be improved by individual titration of all AOCs. This is a major undertaking with a large panel and we chose to use all AOCs at the same concentration in the current panel. We have titrated these jointly and found performance to be good with AOCs around 2-5 ug/mL. We have added a section to discuss this which also Reviewer #2 requested, on page 17.

3. What is the sequencing cost? Can the authors provide some general heuristics about the logistics around this?

Each cell needs 120,000 reads for accurate and robust mapping, which has been added into the manuscript on page 14. This corresponds to about 0.1 USD per cell in sequencing costs depending on instrument and set-up. We'd rather not state the cost of sequencing in the manuscript as this will reduce further over the next few years, but we have stated the number of reads required on page 14.

4. It doesn't appear that there is any validation in the manuscript to determine how close two proteins need to be to be detected as neighbors. Quantitative experiments to determine the radius of interaction for this process is a critical part of this paper and should be added.

In the manuscript, we do not claim to directly measure protein interactions. As mentioned above, it is tricky to precisely determine the MPX resolution. See the new section in Supplementary Figure S17 that discusses DNA pixel size and resolution.

5. How is assay performance impacted by the number of antibodies used for staining? Replicate experiments with smaller subsets of reagents should be performed to quantitatively determine how this impacts assay performance.

When developing the method, we did not observe any deterioration in performance when increasing the panel size from 20 targets to 80 targets. Therefore, we do not anticipate increasing the panel size further will have any negative effect on performance. Importantly, a

few highly abundant targets such as HLA-ABC, B2M, and CD45 should be present in the panel to ensure adequate connectivity of the graphs generated from the data.

Also, various cell types within a PBMC sample will have various types and numbers of AOCs binding, but all cell types provide adequate mapping of each cell with good connectivity.

On page 4 we have now elaborated on this topic.

Decision Letter, first revision:

Dear Simon,

Thank you for submitting your revised manuscript "Molecular Pixelation: Spatial proteomics of single-cells by sequencing" (NMETH-A53085D). It has now been seen by the original referees and their comments are below. The reviewers find that the paper has improved in revision, and therefore we'll be happy in principle to publish it in Nature Methods, pending minor revisions to satisfy the referees' final requests (as described in your recent email to us) and to comply with our editorial and formatting guidelines.

TRANSPARENT PEER REVIEW

Please note: we allow redactions to authors' rebuttal and reviewer comments in the interest of confidentiality. If you are concerned about the release of confidential data, please let us know specifically what information you would like to have removed. Please note that we cannot incorporate redactions for any other reasons. Reviewer names will be published in the peer review files if the reviewer signed the comments to authors, or if reviewers explicitly agree to release their name. For more information, please refer to our FAQ page.

ORCID

Sincerely,
Madhura

Madhura Mukhopadhyay, PhD
Senior Editor

Nature Methods

Reviewer #1 (Remarks to the Author):

Karlsson et al. improved their paper by performing additional experiments, which I appreciate. However, the authors' revision is insufficient to clarify two of my previous concerns.

1. The authors still need solid validation of their methods. The authors claim their methods can identify "the relative locations of many proteins in single-cells" and "spatial resolution in 3D". I still believe that the authors need to compare the fluorescence image with the image obtained by MPX as ref 12 did. If the authors cannot do this experiment, the authors, at least, should give a quantitative estimation of the size of a DNA-Pixel on a fixed cell, which is missing. This value will be considered as the resolution of their methods. The authors still state that DNA-Pixels are smaller than 100 nm in diameter. This gives the impression that the spatial resolution of their methods is 100 nm.
2. As for the colocalization, the authors answered that they detect dynamic changes in colocalization. They provided an MPX colocalization Score. My question was, what is the physical interpretation of the MPX colocalization score? Using a known interaction pair, the authors can do a positive control measurement and may give a quantitative example.

Reviewer #2 (Remarks to the Author):

The authors have presented a revised version of their manuscript on the MPX method here, including much better visual clarification of the approach, a much needed update to the Material & Methods section, and minor experimental aspects.

To this reviewer, the level of rigor applied to the analysis and reagent validation (e.g. antibody titration) also appears to be inadequate, even when additional attempts to describe them were appended in the revision. In its current form, MPX is both low throughput (~500 cells analyzed in Fig 2), single-plex (no concurrent RNA measurements), and has not been shown to work in anything other than cell culture (PBMC or cell lines).

This work would be significantly more impactful over existing technologies if it can overcome at least 2 of the 3 limitations shown above. This revision has not yet been convincing of such. In lieu of the 3 major limitations, should MPX be able to elucidate a significant biological finding due to its novel application, that would serve to convince this reviewer and other readers too.

As such, I unfortunately cannot recommend the manuscript in its current form.

Reviewer #3 (Remarks to the Author):

The authors have addressed all of our questions.

Author Rebuttal, first revision:

Reviewers feedback and our comments on the revised version

We'd like to thank the reviewers and editors for their work on helping improve our manuscript.

Many issues raised by the three reviewers in the first round were similar and most regarding validation of the MPX method. Reviewer 1 and 2 still have some concerns regarding validation which are further clarified in the letter below. Please note that Reviewer 3 had many of the same concerns on method validation, but according to him/her, all these issues have already been adequately addressed in the revised manuscript.

We firmly believe that we have addressed all comments adequately and further clarify our arguments below as well as outline potential modifications to the manuscript.

Best regards,
Simon Fredriksson

Reviewers' Comments:

Reviewer #1:

Remarks to the Author:

Karlsson et al. improved their paper by performing additional experiments, which I appreciate. However, the authors' revision is insufficient to clarify two of my previous concerns.

1. The authors still need solid validation of their methods. The authors claim their methods can identify "the relative locations of many proteins in single-cells" and "spatial resolution in 3D". I still believe that the authors need to compare the fluorescence image with the image obtained by MPX as ref 12 did.

We thank the reviewer for this comment, and agree that validation is vital. In the revision we added Extended Data Fig. 9 containing five new microscopy images of cells where 2 proteins are dispersed (or non polarized) and 3 are polarized, perfectly matching the findings in our MPX data. With these additional validation experiments, all highlighted spatial polarizations are shown in orthogonal immunofluorescence experiments; CD3 (Figure 3D) and CD20/Rituximab (Figure 3H), are shown to polarize in T cells and Raji cells respectively by both MPX and IF, and the additional data includes IF evidence that CD37, CD50, and CD162 polarize and localize to the uropod in T cells, corroborating our previous claims.

It is not technically possible to analyze the exact same cell with both MPX and Microscopy as was done in the Ref 12 for fluorescent proteins and RNA in the same immobilized cells, but our paired experiments provide orthogonal validation of MPX by IF microscopy.

When working with cells in suspension, the traditional validation for single cell work, whether it be scRNA-seq, or of proteins by flow cytometry, is done by mixing two or more cell types as we did in Extended Data Fig. 2 and showing the clear separate detection of such different cells and the correct detection of their respective unique proteins. We appreciate this comment by reviewer #1 but respectfully believe we have thoroughly validated MPX as concluded also by Reviewer #3.

If the authors cannot do this experiment, the authors, at least, should give a quantitative estimation of the size of a DNA-Pixel on a fixed cell, which is missing. This value will be considered as the resolution of their methods. The authors still state that DNA-Pixels are smaller than 100 nm in diameter. This gives the impression that the spatial resolution of their methods is 100 nm.

In the revised manuscript we added a new section in Supplementary Figure S1 discussing the resolution of MPX and made the requested estimation on resolution with an upper limit of 280 nm.

Furthermore, we introduced a disclaimer stating that the estimation from SEM measurements does not necessarily reflect the true resolution of the MPX method. Consequently no actual claims of specific levels of resolution are made in the revised manuscript. We also state that "Further investigations are ongoing to determine a more precise spatial resolution of MPX." But as we have pointed out previously, the molecular nature of the Pixels and the method itself, make it experimentally difficult to accurately determine the resolution and therefore it is reasonable to keep the current best estimation presented and suggest to soften the language to help readers understand the uncertainties therein and include a discussion on resolution in the main text as well, not just in the supplementary part.

2. As for the colocalization, the authors answered that they detect dynamic changes in colocalization. They provided an MPX colocalization Score. My question was, what is the physical interpretation of the MPX colocalization score? Using a known interaction pair, the authors can do a positive control measurement and may give a quantitative example.

We thank the reviewer for this question. However, we do not claim that MPX is able to discern physical protein-protein interaction at its current stage. As such, the MPX colocalization score does not prove physical interaction between two protein molecules in complex, but instead whether they are detected closer to or further apart from one another compared to what is expected by random chance as is stated in the manuscript.

Validating a protein interaction pair with a spatial technology requires a resolution of around 10-20 nm which is below the estimated resolution of MPX but is possible with FRET or Super

Resolution Microscopy, which are on the other hand both very limited in multiplexing and throughput.

While it is conceivable that protein-protein interaction studies could be possible with MPX in the future, given increased resolution and further improved statistical measures, the current iteration of MPX colocalization scores operate at a larger length scales than protein-protein interactions with the ability to detect spatial phenomena such as clusters, polarization, and cell-cell aggregates.

Reviewer #2:

Remarks to the Author:

The authors have presented a revised version of their manuscript on the MPX method here, including much better visual clarification of the approach, a much needed update to the Material & Methods section, and minor experimental aspects.

To this reviewer, the level of rigor applied to the analysis and reagent validation (e.g. antibody titration) also appears to be inadequate, even when additional attempts to describe them were appended in the revision.

In our opinion, antibody titration does not validate the reagent but finds its optimal working concentration. If the antibody (or AOC) is found to bind the correct cell type (for example a B-cell marker on a B-cell, but no signal on a T-cell) it is validated to bind its correct target especially if the particular antibody clone used has been cited in hundreds of previous studies, as most of the clones used in our method have been. Fine tuning an assay by antibody titration can further improve the S/N ratios. We'd be happy to further clarify this in the manuscript, but feel it is already in order.

When working with highly multiplexed immunoassays using antibodies with DNA-tags (AOCs), the individual AOCs are not always titrated. Instead a fixed concentration for all AOCs is often used as in the Proximity Extension Assays for plasma proteomics as described in Assarsson et al 2014 (<https://pubmed.ncbi.nlm.nih.gov/24755770/>). If a particular antibody clone does not behave well at the set fixed universal concentration, it is simply replaced by another clone for the same target. This greatly simplifies the assay development phase, but also presents an opportunity for even further improvement by individual titration if necessary, which we view as a positive note.

In its current form, MPX is both low throughput (~500 cells analyzed in Fig 2), single-plex (no concurrent RNA measurements), and has not been shown to work in anything other than cell culture (PBMC or cell lines).

We are thankful for this comment identifying the great potential to further develop the MPX technology. We agree with the reviewer that these are directions of development that the

method could benefit from. However, we would like to respectfully assert that these points represent significant endeavors that are beyond the scope of this study.

Indeed, MPX currently has lower throughput than scRNAseq or Flow cytometry in the number of cells analyzed, but those are not spatial methods. Currently, microscopy is the golden standard for studying polarization or colocalization of cell surface proteins. In the realm of spatial single-cell proteomics, we would therefore argue that the MPX technology should be compared with microscopy in terms of throughput. Very few (if any) microscopy-based technologies exist that can detect and localize 76 proteins and 4 controls in parallel. Moreover, MPX facilitates mapping the three dimensional distribution of these proteins, which would require an enormous effort with microscopy using iterative staining, bleaching and imaging with Z-stacking.

Furthermore, it is feasible to analyze more cells than we have reported ($n = 1000$), given a higher sequencing capacity. So, MPX throughput in regards to number of cells is more of a sequencing capacity issue.

The term "single-plex" by reviewer #2 is here presumed to mean "single modal" as MPX is highly multiplex (detecting many different molecular species), but is currently only detecting them in one mode (one type of molecule; proteins). Indeed, MPX as presented is concurrently not detecting RNA in the same cells, but it is feasible to do so in the future and we agree with the reviewer that this would be a worthwhile improvement in addition to the spatial protein detection by MPX on single cells. However, this endeavor is outside the scope of this study and a novel method can't reasonably be expected to be multimodal in its first iteration. Furthermore, as we believe we have proven, MPX is unparalleled in its multiplexity and provides an additional layer of information (the spatial dimensions) which distinguishes the technology from other single-cell proteomics assays. Spatial omics methods have garnered significant acclaim in the life sciences over the past decade but are largely focused at studying molecular phenomena that take place at the tissue scale. The MPX technology pushed the resolution to the sub-cellular level, allowing investigation of molecular phenomena at a scale that has remained largely unexplored.

The vast majority of drug targets are proteins and most of the immunotherapy targets by CAR-T or antibodies are cell surface proteins. Flow cytometry for proteins is an invaluable research tool and one would not dismiss its value and impact by the fact that it does not measure RNA simultaneously in each single cell. Similarly, MPX should not be dismissed because it is currently single modal.

Indeed, the first MPX panel is developed for immune cells due to the vast amount of research and drug development done on these sample types and the diagnostic value of analyzing PBMC in for example leukemias. All immuno-oncology in vitro drug development is done on these sample types. It is undoubtedly an interesting direction for further development to develop panels for various sample types, but this application is outside the scope of this study. The fact that this application has not been proven does not dismiss the evidence that MPX works for immune cells which are involved in virtually every disease.

This work would be significantly more impactful over existing technologies if it can overcome at least 2 of the 3 limitations shown above. This revision has not yet been convincing of such. In lieu of the 3 major limitations, should MPX be able to elucidate a significant biological finding due to its novel application, that would serve to convince this reviewer and other readers too.

We agree with the reviewer that the 3 above-mentioned suggestions for increasing the cell throughput, modality, and sample applications are interesting venues to explore for the development of MPX, and that this shows great potential for further development of the technology. However, we believe that the method has great scientific value in its current state, which is bolstered by the fact that MPX is currently being used by many labs around the world generating even more of these requested "significant biological findings" beyond our reported findings on uropod formation in regards to both co-localizations and segregations.

As such, I unfortunately cannot recommend the manuscript in its current form.

Reviewer #3:

Remarks to the Author:

The authors have addressed all of our questions.

We thank Reviewer 3 for all the comments and the time spent reviewing our work manuscript. It is worth noting that we have addressed all the questions raised according to Reviewer #3, which many overlap with Reviewer 1 and 2 and are quoted below from the first review.

- All the major concerns for this work are regarding the validation.
- How difficult is it to optimize this assay? How sensitive is assay performance to antibody titers? - Most cytometric approaches involving surface staining of antibodies require multiple rounds of titration, is this true here?
- It doesn't appear that there is any validation in the manuscript to determine how close two proteins need to be to be detected as neighbors.
- How is assay performance impacted by the number of antibodies used for staining?

Final Decision Letter:

Dear Professor Fredriksson,

I am pleased to inform you that your Article, "Molecular Pixelation: Spatial proteomics of single-cells by sequencing", has now been accepted for publication in Nature Methods. The received and accepted dates will be 01 Sep 2023 and 02 Apr 2024. This note is intended to let you know what to expect from us over the next month or so, and to let you know where to address any further questions.

Over the next few weeks, your paper will be copyedited to ensure that it conforms to Nature Methods style. Once your paper is typeset, you will receive an email with a link to choose the appropriate publishing options for your paper and our Author Services team will be in touch regarding any additional information that may be required. It is extremely important that you let us know now whether you will be difficult to contact over the next month. If this is the case, we ask that you send us the contact information (email, phone and fax) of someone who will be able to check the proofs and deal with any last-minute problems.

Please note that *Nature Methods* is a Transformative Journal (TJ). Authors may publish their research with us through the traditional subscription access route or make their paper immediately open access through payment of an article-processing charge (APC). Authors will not be required to make a final decision about access to their article until it has been accepted. Find out more about Transformative Journals

Authors may need to take specific actions to achieve compliance with funder and institutional open access mandates. If your research is supported by a funder that requires immediate open access (e.g. according to Plan S principles) then you should select the gold OA route, and we will direct you to the compliant route where possible. For authors selecting the subscription publication route, the journal's standard licensing terms will need to be accepted, including self-archiving policies. Those licensing terms will supersede any other terms that the author or any third party may assert apply to any version of the manuscript.

You may wish to make your media relations office aware of your accepted publication, in case they consider it appropriate to organize some internal or external publicity. Once your paper has been scheduled you will receive an email confirming the publication details. This is normally 3-4 working days in advance of publication. If you need additional notice of the date and time of publication, please let the production team know when you receive the proof of your article to ensure there is

sufficient time to coordinate. Further information on our embargo policies can be found here:
<https://www.nature.com/authors/policies/embargo.html>

If you are active on Twitter/X, please e-mail me your and your coauthors' handles so that we may tag you when the paper is published.

Best regards,
Madhura

Madhura Mukhopadhyay, PhD
Senior Editor
Nature Methods